# Characterisation of cell-scale signalling by the core planar polarity pathway during *Drosophila* wing development

Alexandre Carayon, Helen Strutt, David Strutt*

School of Biosciences, University of Sheffield, Sheffield, United Kingdom

## eLife Assessment

This **useful** paper examined the mechanism of planar cell polarity (PCP) using *Drosophila* pupal wing, investigating how 'cellular level', 'molecular level' and 'tissue level' mechanisms intersect to establish PCP. This represents progress for the field and the conclusions are mostly backed up by **solid** data. Whereas the manuscript is sound overall, remaining concerns could be addressed by textual clarification of the concepts used in the manuscript.
[Editors' note: this paper was reviewed by Review Commons and revised by the authors.]

**Abstract** In developing epithelia, cells become planar polarised through asymmetric localisation of the core planar polarity proteins to opposite cell membranes, where they form stable intercellular complexes. Current models differ regarding the signalling mechanisms required for core protein polarisation. Here, we investigate the existence of cell-intrinsic cell-scale signalling in vivo in the *Drosophila* pupal wing. We use conditional and restrictive expression tools to spatiotemporally manipulate core protein activity, combined with quantitative measurement of core protein distribution, polarity, and stability. Our results provide evidence for a robust cell-scale signal, while arguing against mechanisms that depend on depletion of a limited pool of a core protein or polarised transport of core proteins on microtubules. Furthermore, we show that polarity propagation across a tissue is hard, highlighting the strong intrinsic capacity of individual cells to establish and maintain planar polarity.

*For correspondence:
d.strutt@sheffield.ac.uk

Competing interest: The authors declare that no competing interests exist.

## Introduction

Planar polarity (also known as planar cell polarity) describes the coordinated polarisation of cells within the tissue plane and is established by molecular mechanisms that are conserved throughout the animal kingdom (*Hale and Strutt, 2015*). Examples of planar polarised structures are hair follicles in the skin and cilia on epithelia such as lung, while planar polarity also controls polarised cell movements during gastrulation that promote axis elongation and neural tube closure (*Goodrich and Strutt, 2011*; *Devenport, 2016*; *Butler and Wallingford, 2017*; *Davey and Moens, 2017*).

Planar polarity has been extensively studied in the *Drosophila* wing, where each cell is planar polarised and produces a distally oriented trichome (*Adler, 2002*). This depends on the activity of the so-called 'core proteins': six proteins that form polarised intercellular complexes at the proximo-distal cell junctions (*Goodrich and Strutt, 2011*; *Devenport, 2014*; *Harrison et al., 2020*). Core protein complexes on the distal cell membrane are composed of the transmembrane protein Frizzled (Fz) and the cytoplasmic proteins Dishevelled (Dsh) and Diego (Dgo), whereas on proximal cell membranes they contain the transmembrane protein Strabismus (Stbm, also known as Van Gogh [Vang]) and the cytoplasmic protein Prickle (Pk), while the transmembrane protein Flamingo (Fmi, also known as Starry

Night [Stan]) is localised on both apposing cell membranes (*Figure 1A*). This core protein distribution on opposite cell membranes in the hexagonal cells of the wing epithelium results in a characteristic zig-zag localisation pattern (*Figure 1B*). Disruption to activity of any single core protein affects this asymmetric pattern of localisation and trichomes no longer point distally (*Harrison et al., 2020*).

The orientation of planar polarity in the developing *Drosophila* pupal wing is governed by processes occurring at different scales (*Aw and Devenport, 2017*; *Harrison et al., 2020*; *Strutt and Strutt, 2021*). At the tissue level, there are 'global cues', which act to provide an overall direction to planar polarity (*Aw and Devenport, 2017*). The identities of the global cues in the wing are still under investigation, however experimental studies have revealed that planar polarity is initially present in a radial pattern pointing towards the pupal wing margin and is then re-ordered into a proximo-distal pattern as a result of mechanical tension from contraction of the wing hinge that induces polarised cell flows and cell rearrangements (*Aigouy et al., 2010*; *Tan and Strutt, 2025*) (reviewed in *Eaton and Jülicher, 2011*; *Aw and Devenport, 2017*; *Butler and Wallingford, 2017*; *Figure 1C, C'*). The origin of the earlier radial pattern is under debate, with evidence being presented both for and against instructive roles for gradients of secreted Wnt ligands or Fat-Dachsous cadherin expression (e.g. *Ma et al., 2003*; *Matakatsu and Blair, 2004*; *Casal et al., 2006*; *Matakatsu and Blair, 2006*; *Sagner et al., 2012*; *Wu et al., 2013*; *Ewen-Campen et al., 2020*; *Yu et al., 2020*). In addition, experiments in which core pathway activity is only activated at timepoints later than hinge contraction suggest an absence of any proximo-distal polarity cues in the developing wing other than the tissue tension induced by the hinge (*Strutt and Strutt, 2002*; *Strutt and Strutt, 2007*; *Brittle et al., 2022*). Specifically, such 'de novo induction' of pathway activity results in strong swirling patterns of polarity across the surface of the wing that do not respect any specific tissue axis.

The initial polarity bias produced by global cues is then thought to be locally amplified by feedback interactions between core protein complexes on specific cell junctions ('Molecular-scale' [*Figure 1A'*]) (*Usui et al., 1999*; *Axelrod, 2001*; *Strutt, 2001*; *Brittle et al., 2022*). Complexes of the same orientation are suggested to be stabilised by positive feedback interactions, whereas complexes of the opposite orientation are destabilised by negative feedback interactions. This leads to sorting of stable complexes into a uniform orientation on proximal and distal cell membranes (*Harrison et al., 2020*; *Strutt and Strutt, 2021*).

Computational studies support the idea that global cues combined with local feedback interactions are sufficient to establish coordinated planar polarity across a tissue (*Amonlirdviman et al., 2005*; *Le Garrec et al., 2006*; *Fischer et al., 2013*). However, they also predict that in the absence of a global cue, local feedback interactions are insufficient to polarise cells. This stands in contrast to the in vivo studies showing that de novo induction of core pathway activity after the onset of hinge contraction results in swirling patterns of polarity that appear uncoupled from global cues (*Strutt and Strutt, 2002*; *Strutt and Strutt, 2007*; *Brittle et al., 2022*). To explain the production of such de novo patterns of swirling polarity in the absence of global cues, the existence of 'non-local' cell-scale signalling has been suggested (*Meinhardt, 2007*; *Burak and Shraiman, 2009*; *Abley et al., 2013*; *Shadkhoo and Mani, 2019*; *Figure 1B*). In this context, 'cell-scale signalling' describes possible intracellular mechanisms acting to promote core protein segregation to opposite cell membranes. For example, this could be a signal from 'distal complexes' at one side of the cell leading to segregation of 'proximal complexes' to the opposite cell edge, or vice versa.

Despite the theoretical and experimental evidence for its existence, there has so far been no systematic investigation into cell-scale signalling in planar polarity. In this manuscript, we use the establishment of proximo-distal planar polarity in the *Drosophila* wing as a model system to explore possible mechanisms. Based on one suggested scenario (*Meinhardt, 2007*), we tested a 'depletion' model whereby one or more core proteins is present in limiting amounts for cell polarisation, thus acting as a non-local inhibitory signal (*Figure 1B*). To do this, we set up a system in which we can induce core pathway de novo polarisation while varying core pathway gene dosage. We coupled this with the use of tagged Fz that acts as a fluorescent timer to discriminate different Fz populations over time. As an alternative possibility, we assessed whether the previously reported transport of core proteins on microtubules (MTs) (*Shimada et al., 2006*; *Harumoto et al., 2010*; *Matis et al., 2014*) could be acting as a cell-scale signal during polarity establishment (*Figure 1B'*). Finally, we studied polarity propagation across the tissue caused by induction of a boundary of Fz expression, in an attempt to characterise the properties of the putative cell-scale signal.

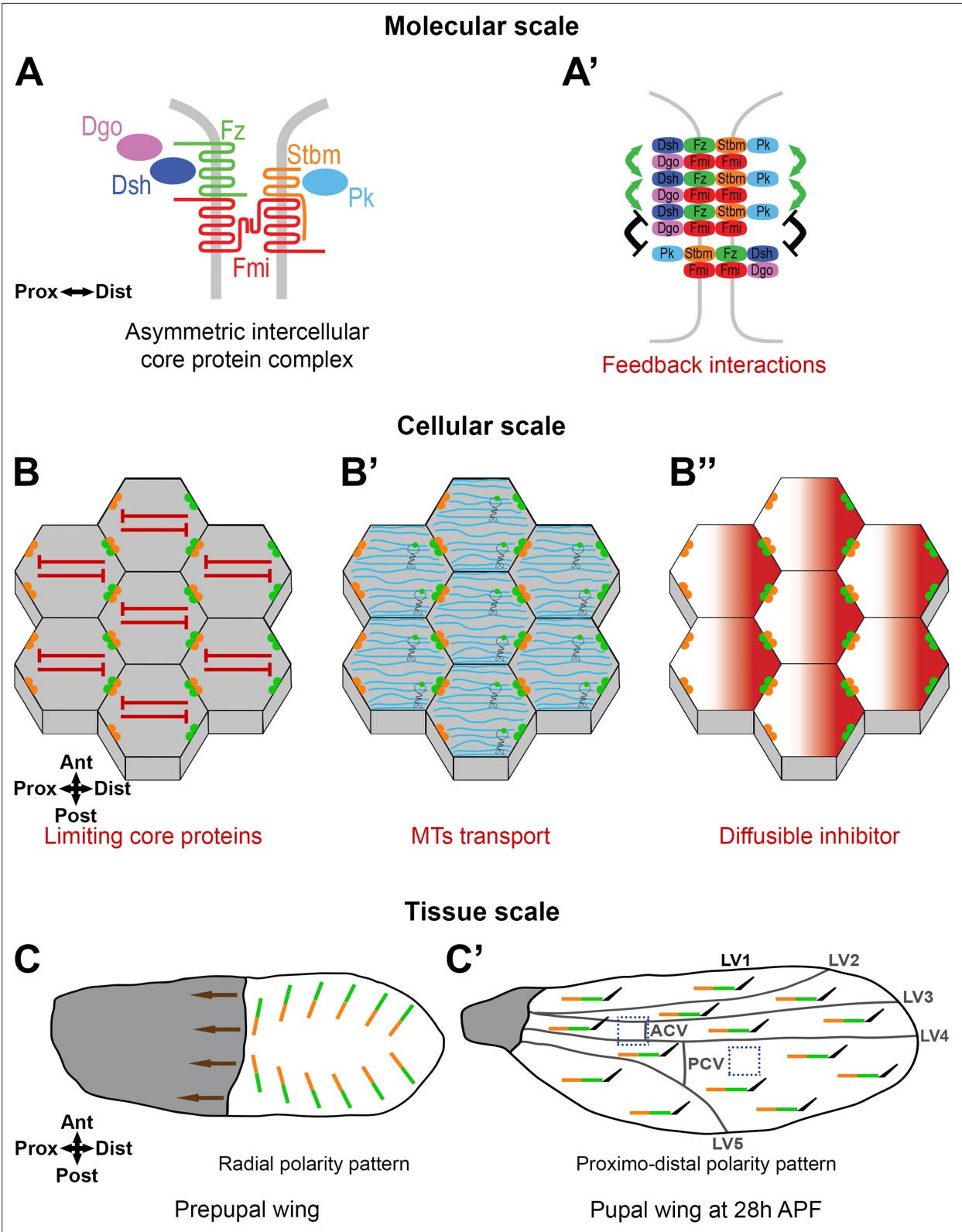

**Figure 1.** Planar polarity in the *Drosophila* wing. (**A**) Core proteins form asymmetric intercellular protein complexes at apico-lateral cell membranes mediating intercellular communication. Frizzled (Fz, green), a seven-pass transmembrane protein, and the cytoplasmic proteins Dishevelled (Dsh, dark blue) and Diego (Dgo, magenta) localise to distal cell membranes, while Strabismus (Stbm, orange, also known as Van Gogh [Vang]), a four-pass transmembrane protein and Prickle (Pk, light blue), a cytoplasmic protein, localise to proximal cell membranes. Flamingo (Fmi, red, also known as Starry

*Figure 1 continued on next page*

*Figure 1 continued*

Night [Stan]), an atypical seven-pass transmembrane cadherin, localises proximally and distally, forming a trans homodimer. (**A′**) Core protein complexes are thought to interact between themselves through feedback interactions locally on cell junctions (Molecular-scale) to form stable clusters, with positive interactions stabilising complexes of the same orientation (green arrows) or negative interactions destabilising complexes of opposite orientation (black symbols). (**B–B″**) Core protein complexes are segregated to opposite cell membranes generating the specific proximo-distal polarised zig-zag core protein localisation pattern which can be promoted by 'Cell-scale signalling'. In this study, we consider several hypotheses to identify such cell-scale signals, such as being mediated through depletion of a limiting pool of a core protein providing long-range inhibition (red symbols) (**B**), through oriented microtubules (MTs) transport (MTs in cyan, with motor proteins) (**B′**), or by a biochemical mechanism such as diffusion of an inhibitor (**B″**). (**C, C′**) In the 28 hr pupal wing, core protein polarity (represented by lines with Stbm in orange and Fz in green) and trichome (black hair) orientation is distal (**C′**). It is suggested that this results from radial global cues generating an initial asymmetric bias across the tissue. During earlier polarity establishment in the prepupal wing, hinge contraction (brown arrows) induces cell rearrangements and a redistribution of the core proteins, from the radial polarity pattern in the prepupal wing (**C**) to aligned core protein polarity along the proximo-distal axis of the pupal wing by 28 hr APF (**C′**).

The online version of this article includes the following source data and figure supplement(s) for figure 1:

**Figure supplement 1.** Tools for Fz induction and measurement of stable fractions.

**Figure supplement 1—source data 1.** zip file containing original uncropped western blots for *Figure 1—figure supplement 1*.

**Figure supplement 1—source data 2.** zip file containing original western blots for *Figure 1—figure supplement 1*, indicating the relevant bands and treatments.

**Figure supplement 1—source data 3.** Excel file containing individual data points for panel D.

Our results argue against roles in mediation of a cell-scale signal for either depletion of a limited pool of a core protein or polarised transport of core proteins on MTs. Nevertheless, our studies of polarity propagation from boundaries reveal that such propagation is hard to induce even over timescales of hours. This suggests that cells possess an intrinsic capacity to polarise and maintain their polarity, implying the existence of a robust cell-scale signal (*Figure 1B″*).

# Results

## Dynamics of de novo planar polarity establishment

To understand mechanisms of core pathway planar polarity establishment, we first followed the process of cell polarisation over time, measuring three parameters: (1) cell polarity orientation (polarity angle), (2) cell polarity magnitude (polarity strength), and (3) core protein complex stability. Cell polarity orientation and magnitude were determined using the principal component analysis (PCA) method in QuantifyPolarity as described previously (*Tan et al., 2021*). Complex stability was assessed by measuring the dynamics of turnover of the core protein Fz. Fz interacts with Fmi to form the asymmetric backbone of core protein complexes (*Strutt and Strutt, 2007*; *Chen et al., 2008*; *Strutt and Strutt, 2008*; *Strutt et al., 2023*) and thus its stability is a proxy for overall complex stability. In our experiments, Fz is tagged with two distinct fluorescent proteins whose maturation timings are different: sfGFP (superfolder GFP) with a very rapid maturation, showing the total population of Fz, and mKate2 with a slower maturation revealing older (presumed stable) populations of Fz, to form a fluorescent timer (*Khmelinskii et al., 2012*; *Barry et al., 2016*; *Ressurreição et al., 2018*; *Figure 1— figure supplement 1A*). The ratio of mKate2/sfGFP is used as a proxy for the proportion of stable Fz.

In wild-type pupal wings, the pattern of core pathway planar polarity depends upon (1) pre-existing polarity established at earlier stages of development, and (2) cell flows and cell re-arrangements that occur during pupal wing morphogenesis (*Aigouy et al., 2010*; *Tan and Strutt, 2025*). To attempt to bypass effects of these upstream polarity cues, we studied planar polarisation in the de novo polarisation context (*Strutt and Strutt, 2002*; *Strutt and Strutt, 2007*; *Brittle et al., 2022*), generated by induction of Fz expression in a *fz* null mutant background at stages of pupal wing development when cell flows and cell re-arrangements are largely complete (see Introduction).

We used the FLP/FRT recombination system (*FRT-STOP-FRT* cassette excision in the presence of *hsFLP*) (*Theodosiou and Xu, 1998*) combined with a 1-hr heat shock (38°C) (*Figure 1—figure supplement 1B*) to induce Fz::mKate2-sfGFP expression under the direct control of the *Actin5C* promoter ubiquitously throughout the wing in a *fz* null mutant background. To confirm that Fz::mKate2-sfGFP had reached steady state levels at our chosen experimental timepoints, we monitored Fz protein by western blot analysis at multiple time points after de novo induction. At 6, 8, and 10 hr after induction, levels were not significantly different from each other, or from our control condition of

constitutive Fz::mKate2-sfGFP levels (*Figure 1—figure supplement 1C, D*). Fz::mKate2-sfGFP levels were several-fold lower than endogenous Fz; however, this does not affect the ability of Fz::mKate2-sfGFP to achieve a normal polarity magnitude (see below). We initially analysed planar polarity establishment in a flat part of the wing, below longitudinal vein 4 (LV 4) and distal to the posterior cross vein (*Figure 1C'*).

To characterise the rate of polarisation and establishment of stable polarity complexes during de novo polarity establishment, we first needed reference measurements for polarised and unpolarised cells. For fully polarised cells, we examined wings that constitutively expressed Act-Fz::mKate2-sfGFP throughout development (*Figure 2D*). Live imaging, with observation of native sfGFP and mKate2 fluorescence, showed Fz asymmetric localisation at cell junctions in the pupal wing, leading to the specific zig-zag core protein localisation pattern (*Figure 2I'–I''*), with a proximo-distal polarity orientation of Fz (*Figure 2I'''*) and a uniform proximo-distal orientation of trichomes in the adult wing (*Figure 2L*), in agreement with previous work (*Strutt, 2001*). Moreover, the magnitude of Fz polarity achieved as measured by the PCA method was similarly to that seen previously for endogenously tagged Fz-EGFP (*Figure 2J*; *Tan et al., 2021*). Hence, mKate2-sfGFP tagged Fz expressed under the *Actin5C* promoter recapitulates endogenous Fz function.

As an unpolarised condition, we used flies deficient for *dsh* planar polarity activity, in which Fz::mKate2-sfGFP was induced de novo for 6 hr in a *fz* null mutant background (polarity induction at 22 hr after prepupa formation [APF] and polarity quantification at 28 hr APF). We characterised this situation as the lowest polarity condition for our study and defined it as 'non-polarised' cells and as having low stability core protein complexes (*Figure 2A, J, K*) consistent with the requirement for Dsh activity for Fz polarisation and stability (*Strutt, 2001*; *Strutt et al., 2011*; *Warrington et al., 2017*). In this context, the specific zig-zag core protein localisation pattern was lost (*Figure 2E'–E''*) and Fz polarity as measured by the PCA method was low (*Figure 2E'', J*).

We then analysed de novo polarity induction in a wild-type background, where expression of Fz::mKate2-sfGFP was induced in a *fz* null background at 6, 8, or 10 hr prior to observation at 28 hr APF (*Figure 2B, F–H*). Induction of Fz::mKate2 for 6 hr revealed detectable planar polarity compared to unpolarised cells (dsh de novo 6 hr vs de novo 6 hr; p = 0.0007) (*Figure 2J*). This polarity increased with induction for 8 and 10 hr to reach a plateau (*Figure 2J*), with a lower peak value than in the control (constitutive Fz::mKate2-sfGFP) condition (de novo 10 hr vs control; p = 0.0003). These three de novo polarity induction conditions gave similar swirling patterns of polarity (*Figure 2F''', G''', H''', M*), as expected for pathway activation in the absence of proximo-distal global cues (*Strutt and Strutt, 2002*; *Strutt and Strutt, 2007*). See *Figure 5—figure supplement 1B–E'''* for further evidence that de novo induction polarity patterns are also not influenced by global cues in the proximal wing.

Quantification of the stable fraction of Fz (mKate2/sfGFP ratio) revealed that core protein complexes from 6 hr after polarity induction are more stable than those in unpolarised cells (dsh de novo 6 hr vs de novo 6 hr; p < 0.0001), reaching a similar level of stability as the normal condition 8 hr after polarity induction (de novo 8 hr vs control; p = 0.4411) (*Figure 2K*). In summary, our findings reveal that following de novo polarity induction in the pupal wing, planar polarity magnitude and complex stability reach a plateau by 8 hr.

## Planar polarity establishment is not highly sensitive to variation in levels of core complex components

After characterisation of de novo polarisation induction in our system, we went on to ask whether de novo polarisation is sensitive to levels of individual core proteins. Based on theoretical considerations suggesting that cell polarisation depends on a cell-scale inhibitory signal (*Meinhardt, 2007*), we hypothesised that the levels of an individual core protein might be limiting for polarisation. This could be due either to levels being limiting for complex formation – thus placing an upper bound on the number of complexes that can form at cell junctions (representing a 'depletion model') (*Figure 1B*) – or due to a dosage sensitive role in inhibitory interactions that sort proteins within the cell (*Figure 1A', B''*).

We therefore looked at the effects of reducing gene dosage by 50% (i.e. the heterozygous condition) during Fz de novo polarisation (6 hr after induction) and once Fz de novo polarisation reaches a plateau (8 hr after induction), and also compared to polarisation in otherwise normal conditions. For *fmi*, *stbm*, *dgo*, and *pk* we used molecularly characterised 'null' alleles (see Materials and methods).

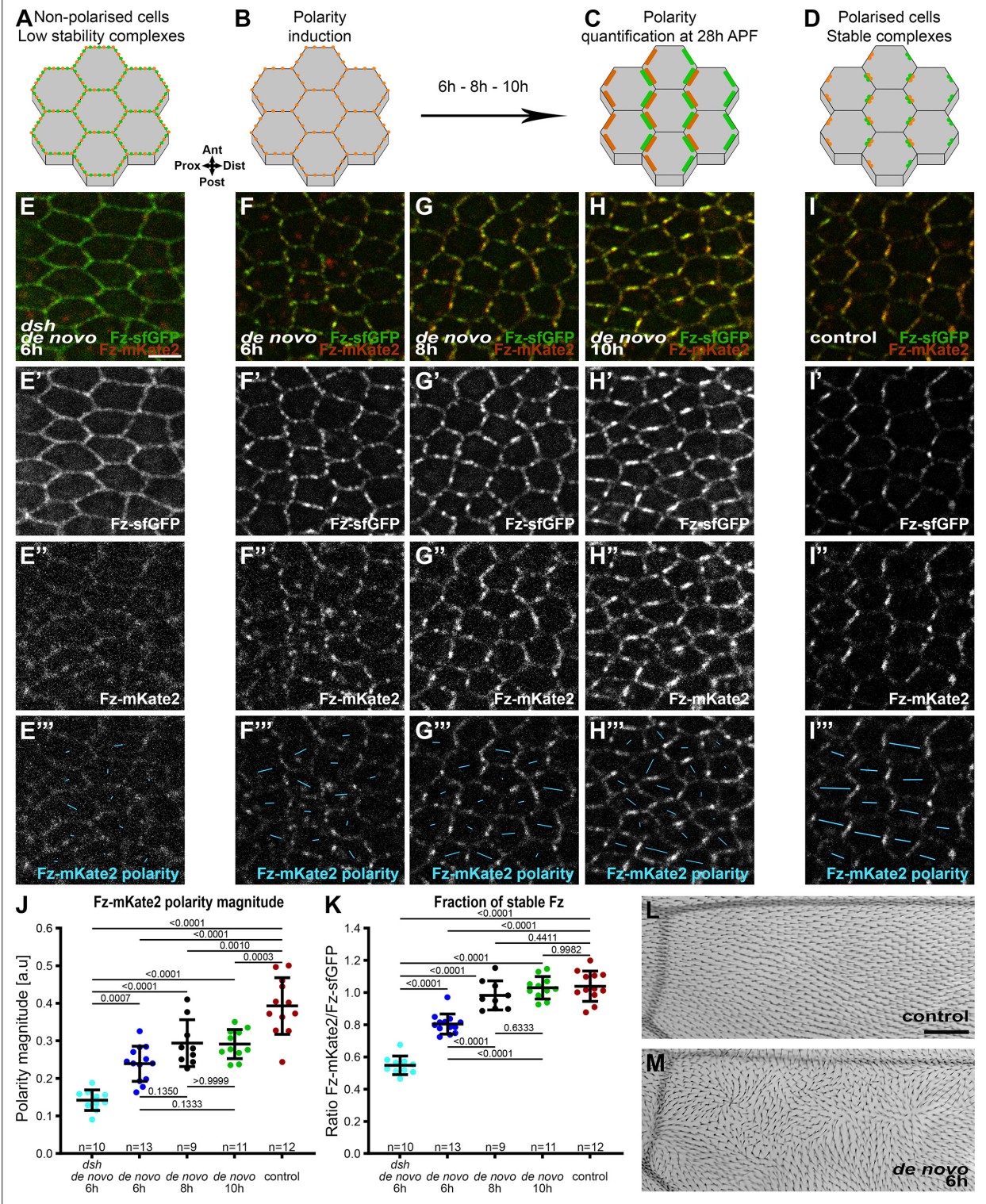

**Figure 2.** Dynamics of planar polarity establishment. Schematics of localisation of core planar polarity pathway protein complexes in cells of the *Drosophila* pupal wing at 28 hr APF, in non-polarised cells (**A**), before polarity induction (**B**), after polarity induction (6, 8, or 10 hr) (**C**), or in the control condition with constitutive expression of Fz::mKate2-sfGFP (**D**), with localisation of Stbm (orange) and Fz (green) at apico-lateral cell junctions indicated. See *Figure 1—figure supplement 1B* for induction timings. (**E–I**) Live confocal images of 28 hr APF pupal wing epithelia expressing Fz::mKate2-sfGFP taken below longitudinal vein 4, see *Figure 1C'* right-hand box. See *Supplementary file 1* for full genotypes. Scale bar, 4 μm and the same hereafter. Native fluorescence for sfGFP (green, **E'–I'**) and mKate2 (red, **E''–I''**), with de novo condition with 6 hr induction in a *dsh¹* mutant background (**E**), with de novo condition with induction for 6 hr (**F**), 8 hr (**G**), and 10 hr (**H**) to establish core protein planar polarity; and in the control condition with constitutive

*Figure 2 continued on next page*

*Figure 2 continued*

expression of Fz::mKate2-sfGFP which gives a normal core protein proximo-distal polarity (**I**). See *Figure 1—figure supplement 1A* for details of Fz tagged to form a fluorescent timer. (**E'''–I'''**) Cell-by-cell polarity pattern of mKate2 fluorescence in pupal wings expressing Fz::mKate2-sfGFP at 28 hr APF. The length and orientation of cyan bars denote the polarity magnitude and angle, respectively, for a given cell. Quantified polarity magnitude based on mKate2 fluorescence (**J**) or fraction of stable Fz as determined by ratio of mKate2/sfGFP fluorescence (**K**), in live pupal wings at 28 hr APF, in the conditions described in (**E–I**). Error bars are standard deviation (SD); *n*, number of wings. ANOVA with Tukey's multiple comparisons test was used to compare all genotypes, p values as indicated. (**L, M**) Images of dorsal surface of mounted adult *Drosophila* wings, below longitudinal vein 4. Scale bar 50 µm. (**L**) Control condition of constitutive Fz::mKate2-sfGFP expression, showing uniform distal orientation of trichomes. (**M**) De novo condition after induction at 22 hr APF, at the equivalent time to the 6 hr induction when imaging at 28 hr APF, showing swirled trichome pattern. Proximal is left, and anterior is up.

The online version of this article includes the following source data for figure 2:

**Source data 1.** Excel file containing individual data points for panel J.

**Source data 2.** Excel file containing individual data points for panel K.

In the case of *dsh*, we used the planar polarity specific *dsh¹* allele to avoid any potential confounding effects of compromising canonical Wg signalling due to Dsh functioning in both pathways (*Axelrod et al., 1998*; *Boutros et al., 1998*). In *dsh¹* mutant tissue, Dsh protein is not seen localised to core protein complexes at cell junctions (*Axelrod, 2001*; *Shimada et al., 2001*) and quantitation of protein distributions supports *dsh¹* being null for planar polarity function (*Warrington et al., 2017*). We have previously shown that the level of Stbm is reduced in the heterozygous condition (*Strutt et al., 2016*). We confirmed that the same was true for Fmi, Pk, and Dgo by carrying out western blot analysis (*Figure 3—figure supplement 1A–D*). The *dsh¹* allele gives rise to a non-functional protein (*Axelrod, 2001*); however, our previous western blot analysis supports Dsh levels being sensitive to gene dosage (*Strutt et al., 2016*).

As we measured Fz stability and polarity using our inducible Fz::mKate2-sfGFP expressing transgene, we were unable to vary Fz dosage in these experiments. Nevertheless, we previously showed that halving *fz* gene dosage did not significantly affect the overall levels of Fz protein (*Strutt et al., 2016*), consistent with Fz production not normally being limiting for core pathway function. Moreover, our western blot analysis indicates that levels of Fz::mKate2-sfGFP are several-fold lower than endogenous Fz levels (*Figure 1—figure supplement 1C, D*), suggesting that our de novo induction conditions most likely represent a 'sensitised' condition for analysing core pathway function.

Notably, we found no significant difference in the degree of polarisation (polarity magnitude) of Fz in polarity complexes at either 6 hr (*Figure 3A–H*) or 8 hr (*Figure 3—figure supplement 2A–H*) after induction, when halving the dosage of *dgo, pk, dsh, stbm,* or *fmi*. Halving the dosage of *pk, stbm,* or *fmi* also had no significant effect on the stable fraction of Fz in polarity complexes at either 6 hr (*Figure 3J*) or 8 hr (*Figure 3—figure supplement 2I*), although a weak decrease was seen in Fz stability in *dsh¹/+* at 6 and 8 hr after polarity induction and in *dgo/+* at 8 hr after polarity induction (*Figure 3J*, *Figure 3—figure supplement 2I*). This is consistent with the known roles of Dsh and Dgo in stabilising Fz at cell junctions (*Strutt et al., 2011*; *Warrington et al., 2017*). Wings homozygous mutant for *dsh* showed the expected low polarity (dsh/dsh de novo 6 hr vs de novo 6 hr; p < 0.0001) (*Figure 3H*) and low stability (dsh/dsh de novo 6 hr vs de novo 6 hr; p < 0.0001) (*Figure 3J*). Interestingly, we did see an effect of core protein gene dosage on the magnitude of Fz polarity achieved (*Figure 3I*), but not on Fz stability (*Figure 3K*) in the control condition of constitutive Fz::mKate2-sfGFP expression. Specifically, halving the gene dosage of *dgo, pk, dsh,* and *stbm* resulted in a reduction in polarity magnitude. This is in contrast to previous results when an effect on asymmetry of endogenous Fmi was not detected after halving gene dosages (*Strutt et al., 2016*). We surmise that this is due to levels of Fz::mKate2-sfGFP expressed constitutively under the *Actin5C* promoter being lower than endogenous Fz levels (*Figure 1—figure supplement 1C, D*), resulting in enhanced sensitivity to reductions in levels of other core proteins.

Overall, our results show that reductions in core protein dosage do not compromise the ability to establish planar polarity during the initial 'fast' phase of de novo polarisation (up to 8–10 hr, *Figure 2J*), despite the lower than normal Fz levels. Fz stability shows a weak sensitivity to levels of functional Dsh and Dgo during early phases of polarisation, but this is not sufficient to affect levels of polarity achieved. However, in wings continuously expressing Fz::mKate2-sfGFP, we do see sensitivity

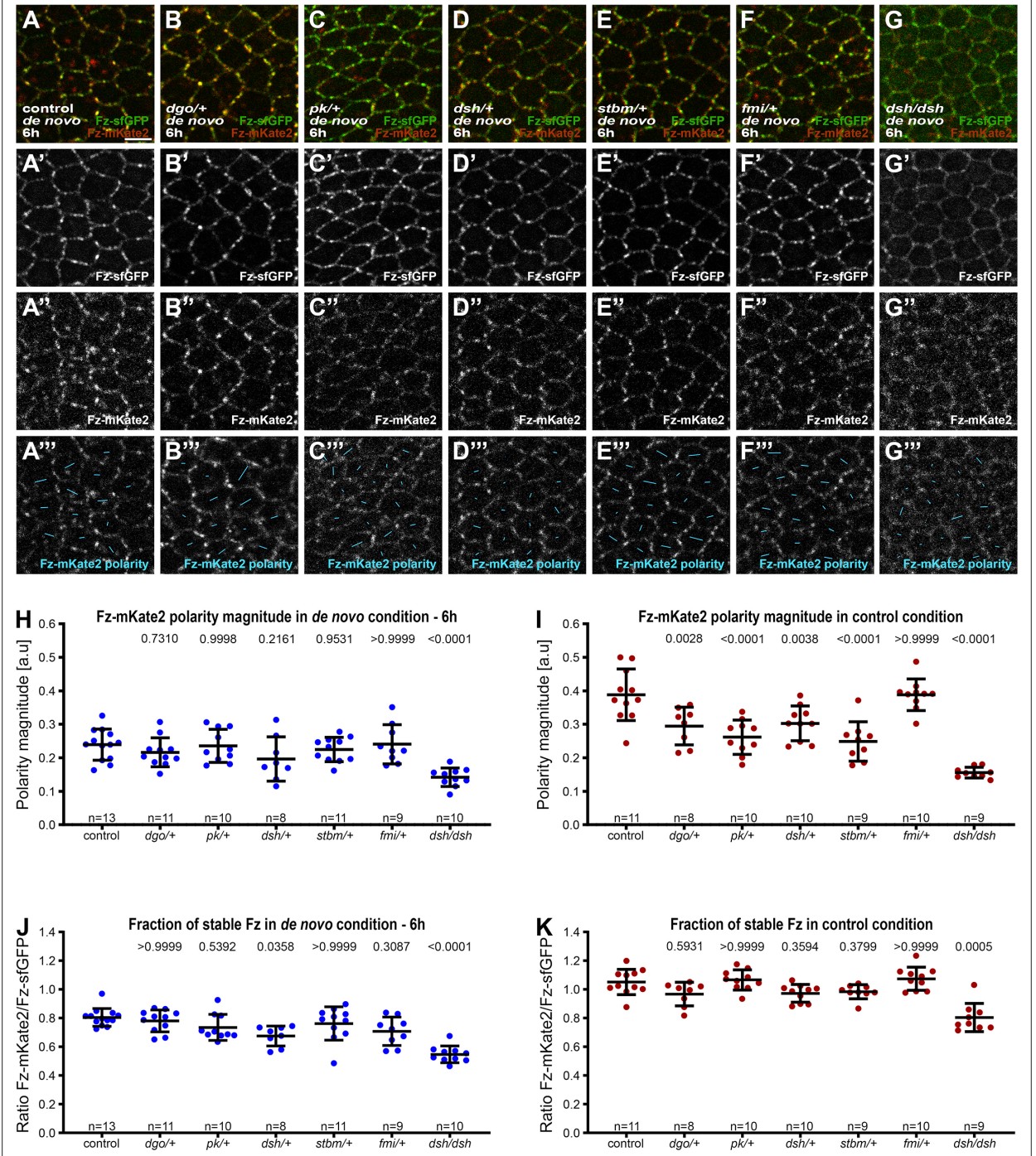

**Figure 3.** Effect of core protein dosage during de novo planar polarity establishment. (**A–G**) Live confocal images of 28 hr APF pupal wing epithelia expressing Fz::mKate2-sfGFP, in de novo condition taken below longitudinal vein 4. Native fluorescence for sfGFP (green, **A'–G'**) and mKate2 (red, **A''–G''**), with de novo condition with 6 hr induction to establish core protein planar polarity in an otherwise wild-type background (**A**), heterozygous mutant backgrounds for *dgo* (**B**), *pk* (**C**), *dsh¹* (**D**), *stbm* (**E**), *fmi* (**F**), and hemizygous mutant background for *dsh¹* (**G**). See ***Supplementary file 1*** for the full genotypes. See ***Figure 3—figure supplement 2*** for live confocal images of equivalent 28 hr APF pupal wing epithelia with constitutive Fz::mKate2-sfGFP expression. (**A'''–G'''**) Cell-by-cell polarity pattern of mKate2 fluorescence in pupal wings expressing Fz::mKate2-sfGFP at 28 hr APF. The length and orientation of cyan bars denote the polarity magnitude and angle for a given cell, respectively. Scale bar, 4 μm. Quantified Fz-mKate2 polarity magnitude based on mKate2 fluorescence (**H, I**) or fraction of stable Fz as determined by ratio of mKate2/sfGFP (**J, K**) in live pupal wings at 28 hr APF, in the conditions described in (**A–G**). Error bars are SD; *n*, number of wings. ANOVA with Dunnett's multiple comparisons tests for quantified Fz-mKate2

*Figure 3 continued on next page*

*Figure 3 continued*

polarity and ANOVA with Kruskal–Wallis multiple comparisons for fraction of stable Fz were used to compare wild-type to mutant backgrounds, p values as indicated. See also *Figure 3—figure supplement 2*.

The online version of this article includes the following source data and figure supplement(s) for figure 3:

**Source data 1.** Excel file containing individual data points for panel H.

**Source data 2.** Excel file containing individual data points for panel I.

**Source data 3.** Excel file containing individual data points for panel J.

**Source data 4.** Excel file containing individual data points for panel K.

**Figure supplement 1.** Decreasing *prickle, flamingo*, and *diego* gene dosage results in lower protein levels.

**Figure supplement 1—source data 1.** zip file containing original uncropped western blots for *Figure 3—figure supplement 1*.

**Figure supplement 1—source data 2.** zip file containing original western blots for *Figure 3—figure supplement 1*, indicating the relevant bands and treatments.

**Figure supplement 1—source data 3.** Excel file containing individual data points for panel A.

**Figure supplement 1—source data 4.** Excel file containing individual data points for panel B.

**Figure supplement 1—source data 5.** Excel file containing individual data points for panel C.

**Figure supplement 2.** Core protein polarity patterns under control conditions and in de novo condition with 8 hr induction to establish planar polarity.

**Figure supplement 2—source data 1.** Excel file containing individual data points for panel H.

**Figure supplement 2—source data 2.** Excel file containing individual data points for panel I.

---

to Dgo, Pk, Dsh, and Stbm levels, consistent with mature polarity being limited when protein levels are reduced.

## MTs do not provide a cell-scale signal for core protein localisation

Our results so far fail to provide evidence that the establishment of 'cell-scale' polarity can be explained by depletion of a limiting pool of a core protein. We therefore decided to explore another possibility, which is that core protein polarity might depend on polarised MT transport (*Figure 1B'*). A role for MTs in mediating an upstream 'global cue' via polarised transport or core proteins has been previously investigated (e.g. *Shimada et al., 2006*; *Harumoto et al., 2010*; *Matis et al., 2014*), but direct functions in core pathway cell-scale polarisation have not been studied. Loss of core protein function has been reported to not alter proximo-distal MT polarity during stages of pupal wing development when core polarity is normally proximo-distal (*Harumoto et al., 2010*); however, this does not rule out the possibility that core proteins help to orient MTs, possibly in parallel to other cues such as cell shape. Such core protein-dependent MT orientation could then amplify cell-scale core protein polarity through polarised core protein trafficking.

To investigate, we used our system of establishing de novo core protein polarity (induction at 22 hr APF and observation at 28 hr APF, in this case expressing Fz-eYFP under control of the *Actin5C* promoter), where core proteins polarise in swirling patterns (*Figure 2F, M*). To follow core protein and MT polarity and cell shape, we fixed and immunolabelled wings and acquired deconvolved images from a flat region of the wing below longitudinal vein 4 (*Figure 1C'*). In wild-type wings, cells were elongated (cell eccentricity close to 0.6) (*Figure 4A*, *Figure 4—figure supplement 1B*) and were aligned following the proximo-distal axis (cell orientation close to 0°) (*Figure 4D*, *Figure 4—figure supplement 1A*). In these cells, both endogenous Stbm protein and MTs were polarised following the same proximo-distal pattern (*Figure 4A"–A'''*, *G, J*, *Figure 4—figure supplement 1C, E*). Without de novo polarity induction (non-induced Fz-eYFP in a *fz* mutant background), there was no change in cell orientation and eccentricity compared to the wild-type condition (*Figure 4B, E*, *Figure 4—figure supplement 1A, B*). However, Stbm was located all around the cell perimeter (*Figure 4B", B'''', H*, *Figure 4—figure supplement 1C, D*), while MTs were still aligned following the proximo-distal axis (*Figure 4B''', K*, *Figure 4—figure supplement 1E*). Induction of de novo polarity did not change cell shape and orientation (*Figure 4C, F*, *Figure 4—figure supplement 1A, B*) but induced a swirling core protein polarity with a partial proximo-distal bias in the studied region below vein 4 (*Figure 4C', C''''*, *I*, *Figure 4—figure supplement 1C, D*), in accordance with the orientation of swirled trichomes in the adult wing (*Figure 2M*). The difference in core protein localisation was not associated with a shift in

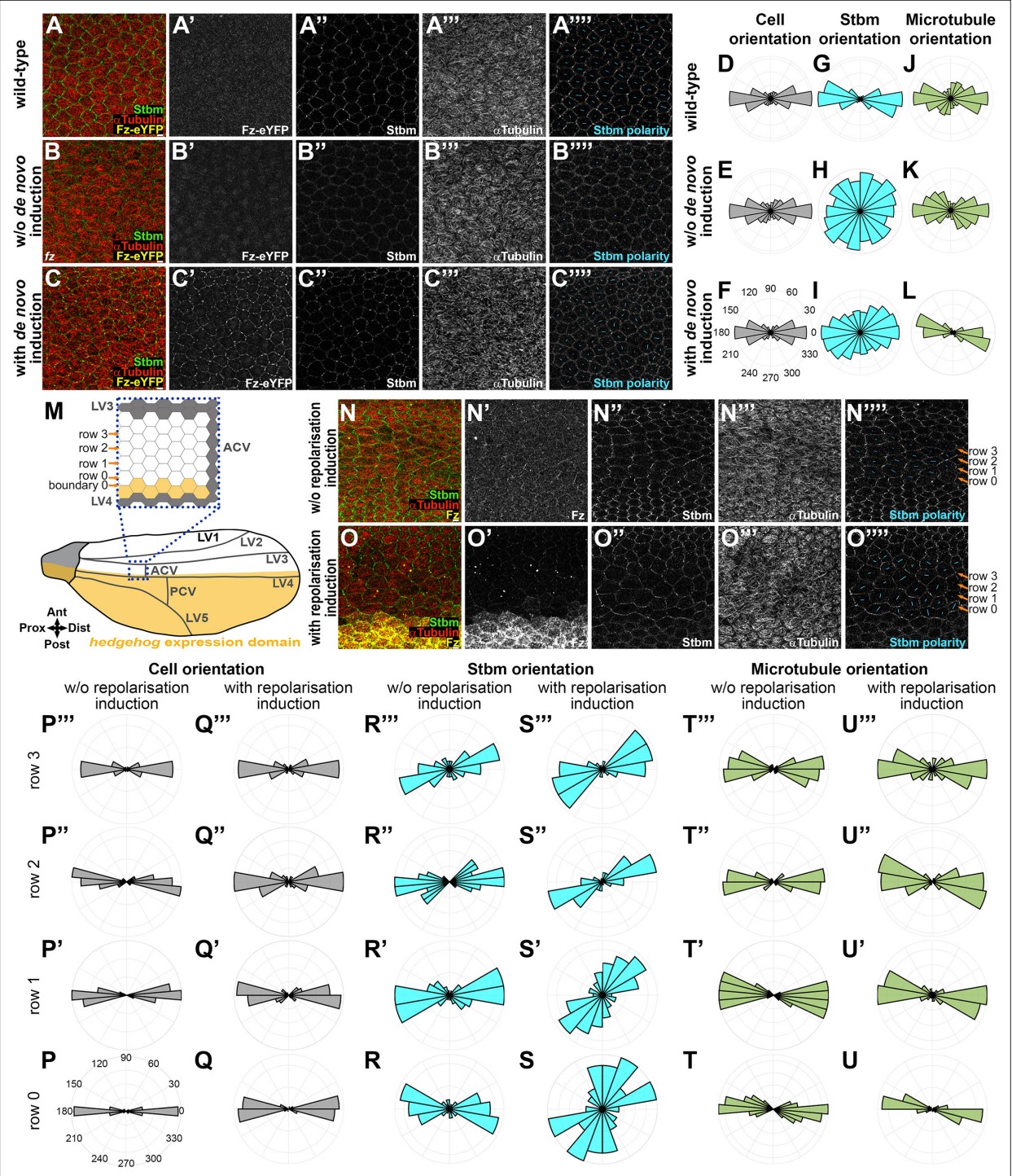

**Figure 4.** Apical microtubules and core proteins are oriented independently of each other. Apical view of cells with deconvolved confocal images of 28 hr APF fixed pupal wing epithelia, distal to the posterior cross vein, below longitudinal vein 4, in a wild-type background (**A**), in de novo Fz polarity establishment condition in a *fz* mutant background without polarity induction (**B**), and with polarity induction for 6 hr (**C**). Native Fz-eYFP fluorescence (yellow, **A', B', C'**), immunolabelling for Stbm (green, **A", B", C"**) and αTubulin (red, **A'", B'", C'"**). Note there is no Fz-eYFP fluorescence at cell junctions detected in **A', B'**. Cell-by-cell polarity pattern of pupal wings in respective conditions (**A"", B"", C""**). The length and orientation of cyan bars denote the polarity magnitude and angle, respectively, for a given cell. See ***Supplementary file 1*** for the full genotypes. See ***Figure 1C'*** for vein locations and imaged wing area. Scale bar, 4 μm. Polar histograms depicting binned cell orientation relative to the horizontal axis (**D–F**), Stbm polarity orientation (**G–I**), and microtubule orientation (**J–L**) relative to the average cell orientation, in wild-type wings (**D, G, J**), in de novo condition without (**E, H, K**) or with (F, I, L) 6 hr Fz-eYFP polarity induction in fixed pupal wings at 28 hr APF. *n* = 7–9 wings for each condition. See also ***Figure 4—figure supplement***

*Figure 4 continued on next page*

*Figure 4 continued*

**1A–E**. (**M**) Cartoon of *hedgehog* expression domain (yellow) in the posterior part of the *Drosophila* pupal wing at 28 hr APF with a zoom in on the proximal region to the anterior cross vein between longitudinal vein 3 and longitudinal vein 4 (blue dash square). Grey cells are vein cells, white cells are intervein cells and yellow cells are intervein cells with *hedgehog* expression. Boundary 0 is between intervein cells expressing or not expressing *hedgehog* and the first row of intervein cells in contact with cells expressing *hedgehog* across boundary 0 is cell row 0 with anteriorly rows 1–3. Apical view of cells from deconvolved confocal images of 28 hr APF fixed pupal wings, in the region proximal to the anterior cross vein between longitudinal vein 3 and longitudinal 4, in an otherwise wild-type background without Fz repolarisation induction (**N**) and with Fz repolarisation induction for 6 hr (**O**). For Fz repolarisation conditions, induced Fz is over-expressed in the *hedgehog* expression domain (posterior wing part) with the UAS-GAL4 system (**M**). Immunolabelling as described in **A–C**, against Fz (yellow, **N′, O′**), Stbm (green, **N″, O″**), and αTubulin (red, **N‴, O‴**). Cell-by-cell polarity pattern of pupal wings in respective conditions (**N″″, O″″**). The length and orientation of cyan bars denote the polarity magnitude and angle, respectively, for a given cell. See **Supplementary file 1** for the full genotypes. See **M** and *Figure 1C′* for vein locations and imaged wing areas. Polar histograms depicting binned cell orientation relative to the horizontal axis (**P, Q**), Stbm polarity orientation (**R, S**), and microtubule orientation (**T, U**) relative to the average cell orientation, without (**P, R, T**) or with (**Q, S, U**) Fz repolarisation induction for 6 hr in fixed pupal wings at 28 hr APF. Cells are grouped in rows relative to their location relative to Fz overexpression (*hedgehog* expression domain) in Fz repolarisation condition or relative to longitudinal vein 4 without Fz repolarisation, with row 0 in contact with the *hh-GAL4* overexpression boundary and row 3 furthest away. n = 8 wings for both conditions. See also *Figure 4—figure supplement 1F–J*.

The online version of this article includes the following source data and figure supplement(s) for figure 4:

**Source data 1.** Excel file containing individual data points for panels D–L.

**Source data 2.** Excel file containing individual data points for panels P–U.

**Figure supplement 1.** Quantification of cell shape, Stbm polarity, and microtubule orientation in wild-type and de novo and repolarisation conditions at 28 hr APF.

**Figure supplement 1—source data 1.** Excel file containing individual data points for panels A–E.

**Figure supplement 1—source data 2.** Excel file containing individual data points for panels F–J.

MT orientation; MTs always followed a proximo-distal orientation (*Figure 4C‴, L, Figure 4—figure supplement 1E*).

To further explore the relationship between core protein polarisation and MT polarity, we next induced core protein repolarisation along the antero-posterior axis. We created an ectopic Fz boundary along horizontal cell membranes (*Figure 4M*) by over-expressing Fz in the *hedgehog* (*hh*) expression domain (posterior compartment) with an inducible UAS/GAL4 system (induction at 22 hr APF and observation at 28 hr APF) comprising *hh-GAL4* and a *UAS-FRT-STOP-FRT-fz* transgene in the presence of *hs-FLP*. We then analysed the region of the wing above this area of protein induction, in the region proximal to the anterior cross vein between longitudinal vein 3 and longitudinal vein 4 where we found that there is the strongest and sharpest boundary of Hedgehog expression (see *Figure 4O′*).

Without induction of core protein repolarisation, in this region of the wing endogenous Fz and Stbm were normally localised along medio-lateral cell junctions (*Figure 4N–N″*), with a proximo-distal polarity orientation (*Figure 4N″″, R–R‴, Figure 4—figure supplement 1H*) as expected. Cells were also oriented along the proximo-distal axis (*Figure 4P–P‴, Figure 4—figure supplement 1F*) with MTs oriented in the same direction (*Figure 4N‴, T–T‴, Figure 4—figure supplement 1J*).

After induction of core protein repolarisation, a Fz overexpression boundary was observable, defining the boundary 0 with cell row 0 (*Figure 4M and O′*). This repolarisation resulted in a relocalisation of Stbm along horizontal cell membranes (*Figure 4O″*), leading to antero-posterior Stbm repolarisation over about one to two cell rows (*Figure 4O″″, S–S″, Figure 4—figure supplement 1H*) and an enrichment of Stbm along boundary 0 due to the Fz overexpression on the opposing boundary (*Figure 4O″, Figure 4—figure supplement 1I*). Despite the reorientation of polarity revealed by the change in Stbm localisation, MT orientation was unchanged, with a proximo-distal orientation in all cell rows (*Figure 4O‴, U–U‴, Figure 4—figure supplement 1J*) and cells oriented along the same axis (*Figure 4Q–Q‴, Figure 4—figure supplement 1F*). We did observe an unexpected slight aspect change towards rounder cells in rows 2 and 3, which are away from the repolarisation boundary (*Figure 4—figure supplement 1G*).

In summary, our results show that core pathway polarity can be established on an orthogonal (antero-posterior) axis with no corresponding change in MT polarity, consistent with the core proteins being able to segregate to opposite cell ends independently of MTs.

## Induced core protein relocalisation on a boundary does not lead to a wave of repolarisation

So far, we have presented evidence against cell-scale polarity relying on a 'depletion' mechanism (i.e. not highly sensitive to core protein levels) or an MT transport mechanism. We have also observed that at the cellular level, over a period of 6 hr an antero-posterior boundary of Fz overexpression can only repolarise core protein localisation for about one to two cell rows (*Figure 4O, S*, *Figure 4—figure supplement 1H*). This short range of repolarisation was unexpected, especially as the same induction regime results in repolarisation of trichomes in the adult wing over at least five rows (*Figure 5C*), consistent with previous reports (*Wu and Mlodzik, 2008*). To investigate further, we examined the sites of trichome emergence in pupal wings at 33 hr APF under similar induction conditions. With this longer induction time, we observed trichomes being repolarised for about three to four cell rows (*Figure 5—figure supplement 1A*), but still less far than observed in adult wings (see Discussion).

We hypothesised that the limited ability of a Fz overexpression boundary to repolarise may be due to three factors. First, there might be a strong and persistent influence of a global cue specifying proximo-distally oriented polarity. Second, there might be strong cell–cell coupling of polarity between neighbours. Hence, a local perturbation, e.g. overexpression of Fz in a neighbouring row of cells, has little effect on the polarity of neighbouring cells, as their polarity is already strongly coupled to their neighbours in the adjacent region of non-overexpression. Third, cells might have a strong intrinsic ability to polarise, while cell–cell coupling of polarity with neighbours might be weak. In this case, once established, polarity is highly robust to the effects of Fz overexpression in neighbouring cells due to relatively weak effects of cell–cell coupling.

In terms of examining these possibilities, we have already attempted to find conditions that weaken (but do not break) the cell-intrinsic polarisation (*Figure 3*) by altering core protein levels, but this was unsuccessful. Moreover, there is no reported way to weaken cell–cell coupling of polarity. Interfering with global cues also presents a challenge, given that they are poorly characterised. However, current evidence suggests that de novo induction at times following the onset of hinge contraction results in a swirling polarity pattern that is not influenced by global cues (see Introduction), a possibility that we investigate further below.

Hence, we decided to compare repolarisation, induced as previously with *hh-GAL4*, in a control condition with constitutive Fz::mKate2-sfGFP expression (where global cues set up proximo-distal polarity), to repolarisation in the de novo condition produced by Fz::mKate2-sfGFP induction at 22 hr APF, where cells are in the process of polarising. Our prediction was that de novo polarity would be easy to orient/repolarise by a boundary of Fz overexpression, as cells (and their neighbours) should not have a pre-existing polarity.

We first looked at the polarity of trichomes in the adult wing. In the control condition of constitutive Fz::mKate2-sfGFP expression, proximal to the anterior cross vein between longitudinal vein 3 and longitudinal vein, as expected, trichomes are oriented along the proximo-distal axis (*Figure 5A*). We found that in this wing region, trichomes also point proximo-distally after de novo polarisation induction of Fz::mKate2-sfGFP expression at 22 hr APF (*Figure 5B*). To investigate the possibility that this proximo-distal polarity in the de novo condition might be due to the presence of a previously unappreciated proximo-distal global cue, we examined the trichome polarity in surrounding regions of the wing. Notably, regardless of whether Fz::mKate2-sfGFP expression was induced at 18, 20, or 22 hr APF, we observed a similar trichome swirling pattern in the proximal wing, with regions proximal, distal, anterior and posterior to the experimental region showing non-proximo-distal adult trichome polarity (*Figure 5—figure supplement 1B–D*). Moreover, quantification of polarity at 28 hr APF in the experimental region showed no difference in the degree of proximo-distal polarisation with longer induction times (*Figure 5—figure supplement 1E*). These observations argue against a proximo-distal global cue being active in this region of the wing at this developmental stage. Instead, we surmise that the proximo-distal polarity in the experimental region is part of the normal trichome swirling pattern in this wing region. Moreover, we can expect to see repolarisation of this proximo-distal polarity to antero-posterior polarity if we overexpress Fz in the posterior compartment.

Interestingly, induction of repolarisation in both the control and de novo conditions resulted in similar degrees of partial trichome repolarisation on the antero-posterior axis (*Figure 5C, D*). In neither case was trichome repolarisation seen beyond longitudinal vein 3. This result does not fit our prediction of longer-range repolarisation in de novo.

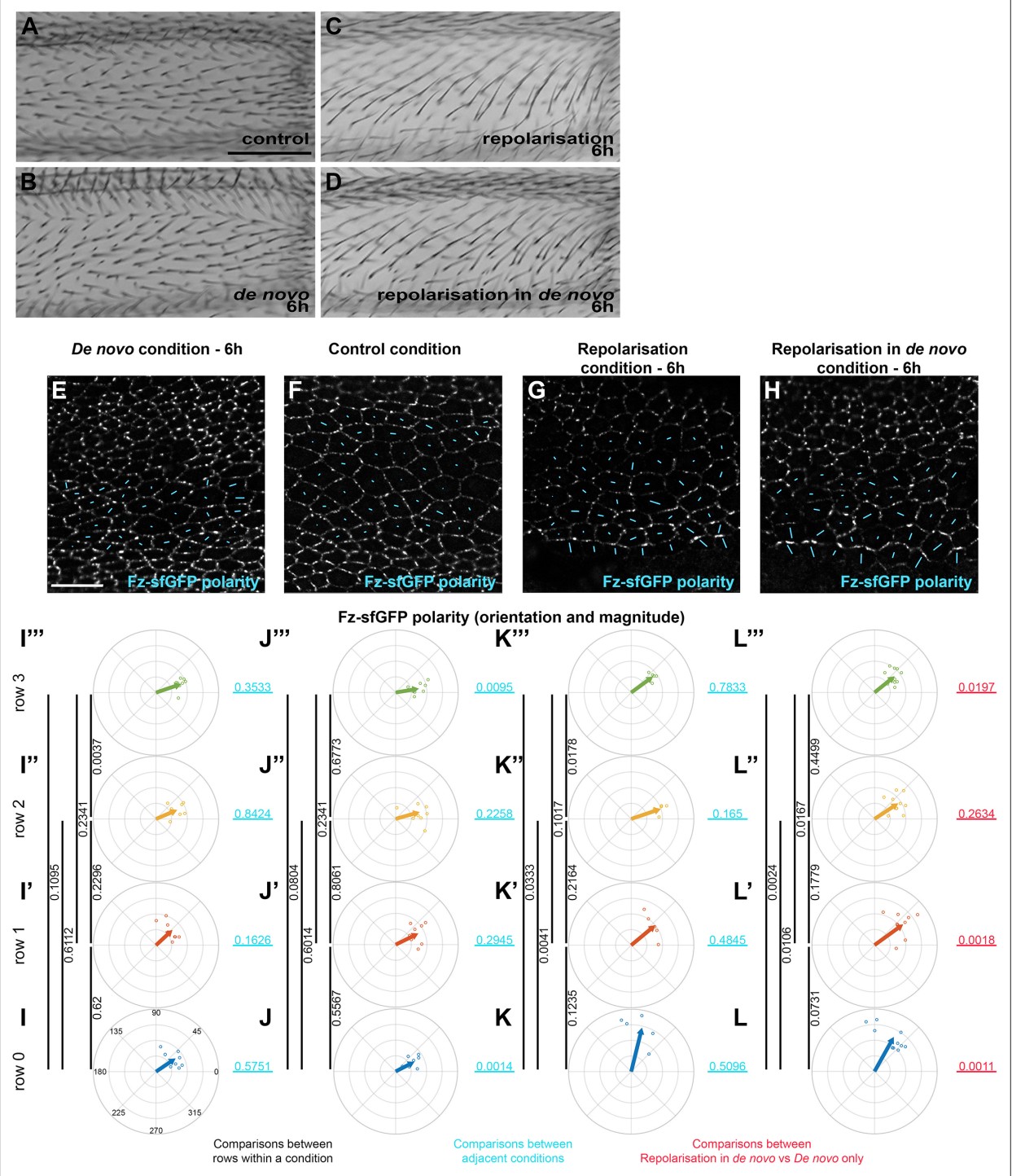

**Figure 5.** Reorientation of planar polarity from a boundary under control and de novo conditions. Images of dorsal surface of adult *Drosophila* wings, taken proximal to the anterior cross vein between longitudinal vein 3 and longitudinal vein 4, showing trichome orientation in control condition with constitutive Fz::mKate2-sfGFP expression (**A**), in de novo Fz::mKate2-sfGFP induction condition (**B**), in repolarisation from the *hh-GAL4* boundary with constitutive Fz::mKate2-sfGFP expression condition (**C**), and in repolarisation from the *hh-GAL4* boundary under de novo Fz::mKate2-sfGFP induction condition (**D**). Repolarisation conditions are under UAS-GAL4 control for Fz overexpression in the posterior wing (*hedgehog* expression domain), with induced Fz expression from *UAS-FRT-STOP-FRT-Fz* in the presence of *hsFLP*. Note, the swirling pattern of polarity generated by de novo induction generates a proximo-distal trichome orientation in this analysed wing area (**B**) but not in surrounding wing regions (***Figure 5—figure supplement 1B–D***). Trichome polarity is reoriented in the antero-posterior direction in both repolarisation conditions. See ***Figures 4M and 1C'*** for vein locations and imaged wing area. See ***Supplementary file 1*** for the full genotypes. Scale bar, 50 μm (**E–H**) Planar polarity measurement at the

*Figure 5 continued on next page*

*Figure 5 continued*

cellular scale in the region proximal to the anterior cross vein, between longitudinal vein 3 and longitudinal vein 4 in fixed pupal wings at 28 hr APF. The length and orientation of cyan bars denote the polarity magnitude and angle for a given cell, respectively. De novo condition with 6 hr induction of Fz::mKate2-sfGFP (**E**), constitutive expression of Fz::mKate2-sfGFP (**F**), repolarisation condition with constitutive expression of Fz::mKate2-sfGFP (**G**), and repolarisation condition with 6 hr de novo induction of Fz::mKate2-sfGFP (**H**). In repolarisation conditions, induced Fz is over-expressed in the posterior wing, whereas in the anterior wing only Fz::mKate2-sfGFP is expressed. In the posterior wing, the two Fz populations (over-expressed Fz and Fz::mKate2-sfGFP) compete for the same membrane locations and Fz::mKate2-sfGFP signal is not evident. Scale bar, 10 µm (**I–L**) Circular plots of quantified total Fz (Fz::sfGFP) magnitude and orientation relative to horizontal axis in the region proximal to the anterior cross vein, between longitudinal vein 3 and longitudinal vein 4 at 28 hr APF in fixed pupal wings. Small dots show polarity angle and magnitude for individual wings, arrows show average polarity and magnitude across all wings. Cells are grouped in rows relative to their location relative to the Fz overexpression domain in Fz repolarisation condition or relative to longitudinal vein 4 without Fz repolarisation, with row 0 in contact with Fz overexpression boundary and row 3 furthest away. (**I–I‴**) de novo condition with 6 hr to establish core protein polarity, (**J–J‴**) control condition with constitutive Fz::mKate2-sfGFP expression, (**K–K‴**) repolarisation condition for 6 hr to re-orient core protein polarity, and (**L–L‴**) repolarisation under de novo condition for 6 hr induced Fz::mKate2-sfGFP expression; in row 0 (**I–L**), in row 1 (**I′–L′**), in row 2 (**I″–L″**), and in row 3 (**I‴–L‴**). Vertical black lines associated with p values on right of each column represent comparisons between different rows of the same polarisation condition. Horizontal pale blue lines associated with p values represent comparison between the two adjacent polarisation conditions for the same cell row. On the far right, horizontal red underlined p values represent comparison between repolarisation in de novo condition versus de novo condition (far left) for the respective rows of cells. Hotelling's *T*-square tests were used to compare total Fz polarity (orientation and magnitude).

The online version of this article includes the following source data and figure supplement(s) for figure 5:

**Source data 1.** Excel file containing individual data points for panels I–L.

**Figure supplement 1.** Trichome and core protein polarity in normal and de novo conditions.

**Figure supplement 1—source data 1.** Excel file containing individual data points for panel E.

We then compared Fz distribution in this region of pupal wings at 28 hr APF in the same four conditions: control constitutive Fz::mKate2-sfGFP expression, 6 hr of de novo Fz::mKate2-sfGFP polarity establishment, 6 hr of repolarisation caused by Fz overexpression in the posterior compartment, and 6 hr of de novo polarity with 6 hr of repolarisation. As we saw in the adult wing, de novo polarity was broadly proximo-distal and similar in magnitude to the control polarity establishment in this region (***Figure 5E, F, I–J‴***).

Surprisingly, in both the repolarisation (***Figure 5G, K–K‴***) and repolarisation in de novo conditions (***Figure 5H, L–L‴***), only row 0 was strongly repolarised (***Figure 5J vs K*** p = 0.0014, ***Figure 5L vs I*** p = 0.0011). In the repolarisation only condition, there was no significant change in polarity in row 1 or 2 (***Figure 5J′ vs K′*** p = 0.2945, ***Figure 5J″ vs K″*** p = 0.2258) although unexpectedly row 3 showed a significant displacement from proximo-distal polarity (***Figure 5J‴ vs K‴*** p = 0.0095). In the repolarisation in de novo condition, row 1 was modestly repolarised towards the antero-posterior axis (***Figure 5L′*** vs ***5I′*** p = 0.0018), but again row 2 showed no detectable repolarisation (***Figure 5L″*** vs ***5I″*** p = 0.2634) and unexpectedly row 3 also showed antero-posterior displacement (***Figure 5L‴*** vs ***5I‴*** p = 0.0197).

The strong repolarisation of row 0 (in cells touching Fz overexpressing cells) is expected, as Stbm is known to be strongly recruited to cell boundaries with high apposing levels of Fz (***Bastock et al., 2003***). However, the variable pattern of weak repolarisation between rows 1 and 3 indicates a failure of this strong repolarisation to propagate from cell to cell, and we suspect the variation may be simply due to sampling noise. In particular, these data indicate that there is no dramatic increase in repolarisation from a boundary in de novo as compared to repolarisation in the control condition, with only a modest difference in polarity in row 1 in the de novo condition.

Overall, our results show that it is hard to repolarise from a boundary of Fz overexpression in both control and de novo polarity conditions, consistent with de novo polarity being rapidly and robustly established and not easily perturbed. This provides evidence in favour of an effective cell-intrinsic polarisation mechanism.

## Cell-intrinsic polarity is robust against local perturbation

Given the different conditions, the similarity of degree of repolarisation in control versus de novo conditions was puzzling. We reasoned that there might be differences in these conditions that were masked by simply using measures of overall cell polarity. We therefore carried out a more detailed analysis, measuring levels of proteins on individual cell junctions (***Figure 6A***).

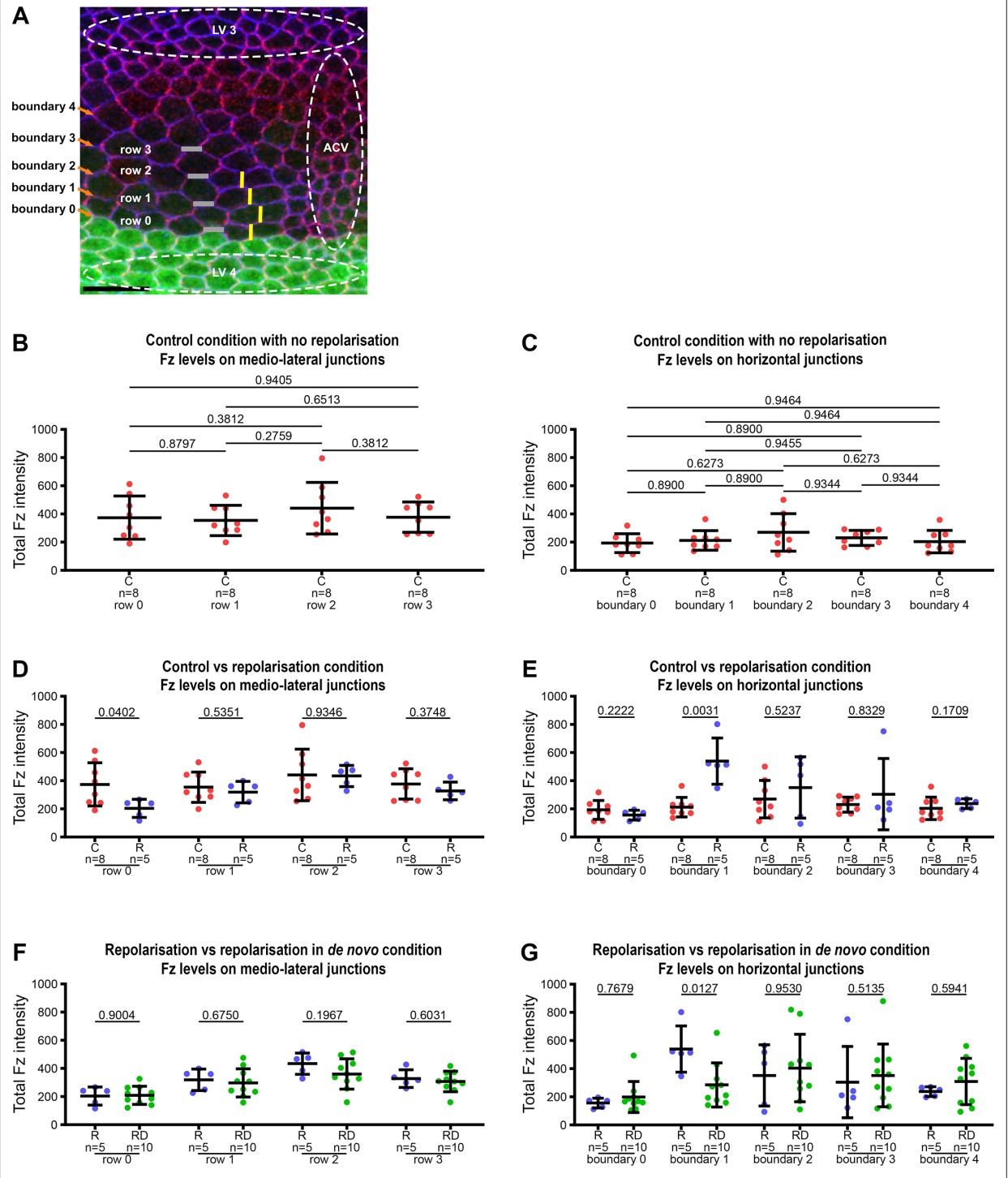

**Figure 6.** Distribution of Fz along medio-lateral and horizontal junctions in Fz repolarisation and repolarisation under de novo conditions. (**A**) Explanatory diagram for wing region, cell rows and cell boundaries. Visualisation of the anterior part of the anterior cross vein (ACV) between the longitudinal vein 3 (LV 3) and the longitudinal vein 4 (LV 4) in a fixed pupal wing at 28 hr APF. Green area indicates the expression domain of the *hh-GAL4* driver used to over-express Fz. E-Cadherin staining is in blue and Stbm staining in magenta. Junctions with horizontal orientation are schematised with grey lines, forming horizontal boundaries with boundary 0 between last row of cells over expressing Fz in LV 4 and the first row of cells without over expression of Fz in the intervein area (row 0). Medio-lateral junctions are schematised in yellow. As in *Figure 5*, repolarisation conditions are under UAS-GAL4 control for Fz overexpression in the posterior wing (*hedgehog* expression domain), with induced Fz expression from *UAS-FRT-STOP-FRT-Fz* in the presence of *hsFLP*. Scale bar, 10 µm. (**B–G**) Quantitation of Fz intensity in fixed pupal wings at 28 hr APF, binned by row of cells relative to their location relative to Fz overexpression in Fz repolarisation and Fz repolarisation under de novo conditions, or relative to longitudinal vein 4 without

*Figure 6 continued on next page*

*Figure 6 continued*

Fz repolarisation induction, with row 0 in contact with Fz overexpression boundary and row 3 furthest away. Boundary 0 is the posterior horizontal junction of row 0 in contact with Fz overexpression area and boundary 1 is the anterior horizontal junction shared between cells in row 0 and row 1. See ***Supplementary file 1*** for the full genotypes. Error bars are SD; *n*, number of wings, p values as indicated. Total Fz intensity in control condition with constitutive Fz::mKate2-sfGFP expression, along medio-lateral junctions (**B**) and along horizontal junctions (**C**). ANOVA with Holm–Sidak's multiple comparisons test was used to compare all conditions. Comparison of Fz distribution in control condition with constitutive Fz::mKate2-sfGFP expression versus repolarisation condition with 6 hr induction, along medio-lateral junctions (**D**) and along horizontal junctions (**E**). Unpaired *t*-test (**D**) and Mann–Whitney test (**E**) were used to compare fluorescence intensities. Comparison of Fz distribution in repolarisation condition versus repolarisation in de novo condition with 6 hr induction, along medio-lateral junctions (**F**) and along horizontal junctions (**G**). Unpaired *t*-test (**F**) and Mann–Whitney test (**G**) were used to compare fluorescence intensities. See also ***Figure 6—figure supplement 1***.

The online version of this article includes the following source data and figure supplement(s) for figure 6:

**Source data 1.** Excel file containing individual data points for panels B–G.

**Figure supplement 1.** Distribution of Fz along medio-lateral and horizontal junctions in repolarisation and repolarisation under 6 and 10 hr de novo induction conditions.

**Figure supplement 1—source data 1.** Excel file containing individual data points for panels A–D.

In control conditions, as expected, all cells in rows 0–3 have similar protein levels on their medio-lateral junctions (***Figure 6B***, yellow lines in ***Figure 6A***), and on their horizontal junctions (***Figure 6C***, grey lines in ***Figure 6A***), with ~2x higher levels on the medio-lateral junctions, consistent with the observed proximo-distal polarity.

After 6 hr repolarisation in the control condition, Fz distribution is only significantly altered on the row 0 medio-lateral junction (***Figure 6D***) and the boundary 1 horizontal junction between rows 0 and 1 (***Figure 6E***). Specifically, Fz is lost from the medio-lateral junctions of cells in row 0 and increased on the horizontal junctions along boundary 1. This is consistent with the changes in cell polarity seen in rows 0 and 1 (***Figure 5G, K–K'''***), and supports the view that polarity changes only propagate one to two rows from a boundary of Fz overexpression.

We then investigated 6 hr repolarisation under de novo conditions. Interestingly, Fz levels along junctions in repolarisation in de novo do not exactly mirror those in repolarisation in the control condition. While Fz levels are similarly low on medio-lateral junctions in row 0 (***Figure 6F***), they are less high on the horizontal boundary 1 between rows 0 and 1 than in repolarisation in normal condition (***Figure 6G***). We questioned whether this difference might be due to the short period of polarisation, but obtained the same result after 10 hr induction (***Figure 6—figure supplement 1A–D***).

Taken together, our findings reveal that induction of repolarisation from a boundary leads to a new core protein distribution along cell membranes, with relocalisation from medio-lateral junctions to horizontal junctions. However, these core protein relocalisations do not induce polarity rearrangement in neighbouring cells. This failure of repolarisation to propagate beyond one to two rows of cells is consistent with the presence of an effective cell-intrinsic polarisation machinery, which is only weakly affected by cell–cell coupling of polarity to neighbours.

## Discussion

The establishment of coordinated planar polarity across a tissue depends on processes acting at different scales (***Devenport, 2014***; ***Aw and Devenport, 2017***; ***Butler and Wallingford, 2017***; ***Harrison et al., 2020***; ***Strutt and Strutt, 2021***; ***Brittle et al., 2022***). These include global cues at the tissue level, cell-intrinsic polarisation mechanisms that can amplify polarity either on cell–cell boundaries or at the level of whole cells, and cell–cell coupling of polarity between neighbouring cells (see ***Figure 1***). An important question is the relative influence of each of these processes on the final polarity pattern.

In this work, we have focused on understanding how cell-intrinsic 'cell-scale signalling' might occur in the *Drosophila* pupal wing, promoting segregation of the core proteins to opposite cell ends. We used several tools including: (1) de novo induction of core pathway planar polarity, to bypass earlier effects of global cues; (2) manipulation of core protein gene dosage to test 'depletion' models for cell-scale patterning; (3) induced repolarisation of planar polarity from a boundary to challenge the effects of cell-intrinsic polarisation mechanisms; and (4) use of a fluorescent timer to monitor Fz stability simultaneously at multiple cell junctions over time.

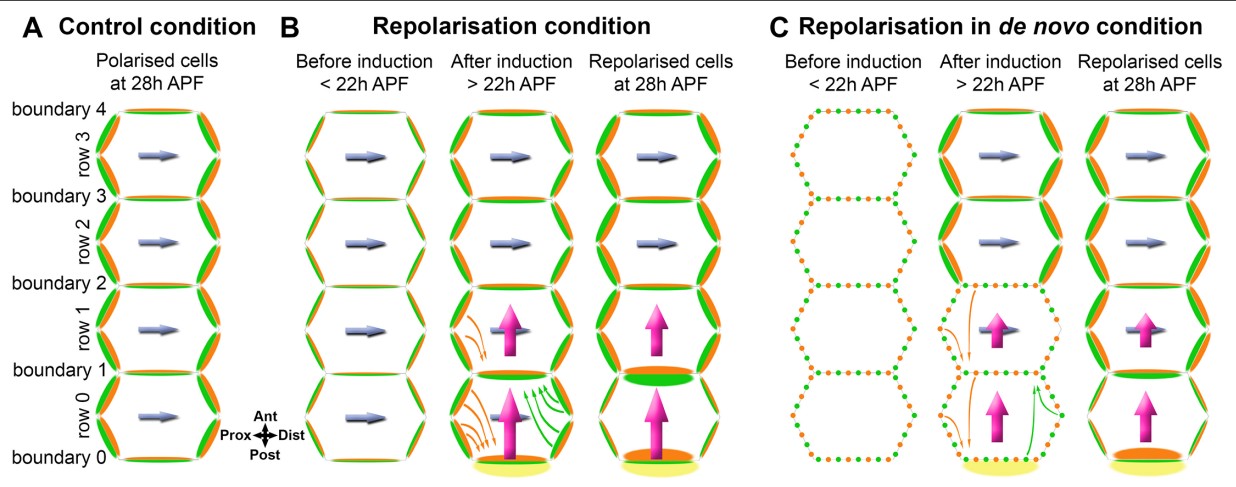

**Figure 7.** Summary model of polarity propagation confronted with cell-scale polarity. (**A**) Core protein polarisation in control (wild-type) conditions at 28 hr APF. The level of total Fz (green) does not vary significantly on different medio-lateral junctions or on different horizontal junctions. Stbm (orange) levels mirror Fz levels on the same cell junctions. Planar polarity is directed by the integrated effects of an upstream extrinsic global cue and a cell-intrinsic cell-scale polarisation system (blue arrow) with a proximo-distal orientation. (**B**) Repolarisation of core protein polarity from a boundary. Before repolarisation induction, Fz (green) and Stbm (orange) become polarised following the proximo-distal axis due to the normal extrinsic and intrinsic planar polarity systems (blue arrow). After repolarisation induction by Fz overexpression in the posterior part of the wing, over-expressed Fz (yellow) accumulates on posterior boundary 0. In row 0, Fz is seen to be lost from medio-lateral junctions and to accumulate on the horizontal junction away from the region of overexpression (boundary 1). This could be due to redistribution of existing protein via endocytosis and trafficking (green arrows) or removal and degradation of protein from medio-lateral junctions and net delivery of new protein to horizontal junctions. Stbm redistribution also occurs (orange arrows) with a loss of Stbm from medio-lateral junctions and accumulation on anterior boundary 0. We surmise that Stbm redistribution to boundary 0 is primarily a result of Stbm being sequestered by forming asymmetric complexes with the high levels of over-expressed Fz on the other side of the boundary. In row 1, it is likely that a loss of Stbm located on medio-lateral junctions, and relocation along boundary 1, again occurs via the same potential mechanisms with sequestration into boundary 1 due to accumulation of Fz on the other side of this boundary. We hypothesise that Stbm redistribution to boundaries 0 and 1 results in overall cell repolarisation of rows 0 and 1 through generation of a cell-scale cell-intrinsic polarity signal (magenta arrows) that overrides the original cell polarity cues (blue arrows). Notably, repolarisation does not propagate into row 2, presumably due to weak cell–cell coupling of polarity and the strong cell-intrinsic polarity mechanisms in these cells (blue arrows). (**C**) Repolarisation of core protein polarity from a boundary in de novo conditions. As in repolarisation of core protein polarity in the control conditions, we surmise that Stbm accumulates on boundary 0 in row 0, due to high Fz levels on the other side of the boundary causing Stbm to be sequestered into asymmetric complexes. This could then result in a cell-scale polarity signal (magenta arrows) on the antero-posterior axis that promotes Fz accumulation on boundary 1 in row 0, rather than on medio-lateral boundaries. Concurrently, cells in rows 1–3 spontaneously polarise and adopt a proximo-distal polarity due to the presence of a strong cell-intrinsic polarisation system (blue arrows). Weak cell–cell coupling of polarity means that the altered polarity in row 0 does not propagate significantly.

Based on experiments in which we attempted to repolarise tissue from a boundary of Fz overexpression, we find support for the existence of a robust cell-scale signalling mechanism that is able to establish and maintain polarity in the face of local perturbations. However, our results fail to provide evidence that such cell-scale signalling is mediated by either a depletion of a limiting pool of an individual core protein, or by reorientation of the apical MT network by core pathway activity.

While we have yet to elucidate the underlying mechanism of cell-scale signalling, we believe that such signalling underpins the reported ability of both insect and mammalian epithelial cells to spontaneously planar polarise with only locally coordinated polarity that does not align with a global tissue axis (*Strutt and Strutt, 2002*; *Strutt and Strutt, 2007*; *Vladar et al., 2012*; *Aw et al., 2016*). Moreover, the conservation across species suggests that there may be a single universal mechanism. As discussed in previous work (*Meinhardt, 2007*; *Burak and Shraiman, 2009*; *Abley et al., 2013*; *Shadkhoo and Mani, 2019*), such a signal may be mediated by diffusion of a cytoplasmic protein (*Figure 1B″*) and might be generated by (for instance) post-translation modification of a core complex component. In this context, it is interesting to note that multiple post-translational modifications of core proteins have been reported (reviewed in *Harrison et al., 2020*), which represent potential candidate mechanisms.

A caveat to our conclusion that cell-scale signalling does not depend on depletion of a limited pool of one of the core proteins is that we only lowered protein levels by 50%. However, we believe

that such conditions would be sufficient to reveal a role for two reasons. First, we assayed the rate of de novo polarisation, rather than simply the final polarity achieved. Our reasoning was that if a core protein were limiting, halving the levels would minimally be expected to reduce the rate of the polarisation process. Second, the experiments were carried out under conditions where Fz levels were several-fold lower than normal, which we would expect to further sensitise the system if one of the other core proteins were limiting. However, we note that we cannot eliminate the possibility that Fz itself is the limiting factor.

Interestingly, repolarisation from a boundary in the control and de novo conditions exhibited different effects on Fz distribution. In the control condition (constitutive expression of Fz::mKate2-sfGFP), repolarisation in row 0 occurs via loss of Fz from medio-lateral junctions and accumulation of Fz on the horizontal junction away from the region of overexpression (*Figure 7B*). This could be either a redistribution of existing protein via endocytosis and trafficking, or removal and degradation of protein from medio-lateral junctions in parallel with delivery of new protein to the horizontal junction (boundary 1), as previously suggested (*Butler and Wallingford, 2017*; *Warrington et al., 2017*). These results are consistent with a cell-scale signal possibly mediated by ectopic recruitment of Stbm to boundary 0, that promotes Fz removal from medio-lateral junctions and localisation to the opposite horizontal junction. In contrast, under de novo conditions, a Fz expression boundary results in repolarisation in row 0 that is characterised by low Fz on medio-lateral boundaries, but no strong Fz accumulation on boundary 1 (*Figure 7C*). Failure to see high Fz on any of the boundaries in row 0 was unexpected, but again supports the presence of a cell-scale signal coming from boundary 0 preventing Fz from accumulating on medio-lateral junctions.

An unexplained observation is that in our repolarisation from a boundary experiments, when observing trichome initiation in the 33 hr pupal wing, we only detect significant repolarisation over about three to four cell rows, but in the adult wing, repolarisation extends at least five cell rows. We propose that there may exist some degree of coupling of polarity between mature trichomes that is not yet present when prehairs emerge at the cell edge (*Wong and Adler, 1993*), which might be mechanical in nature, or be mediated for instance by septate junction proteins (*Venema et al., 2004*).

In conclusion, this work represents the first attempt to systematically study cell-scale signalling in core pathway planar polarity. We provide evidence for and against different possible mechanisms, providing the basis for future work.

## Materials and methods

**Key resources table**

| Reagent type (species) or resource | Designation | Source or reference | Identifiers | Additional information |
|---|---|---|---|---|
| Antibody | Mouse monoclonal anti-α-Tubulin DM1A | Sigma | cat#T9026 ; RRID:AB_477593 | (1:500 for immunofluorescence) |
| Antibody | Rabbit polyclonal anti-Stbm | *Warrington et al., 2013* | N/A | (1:1000 for immunofluorescence) |
| Antibody | Rat anti-Stbm | *Strutt and Strutt, 2008* | N/A | (1:1000 for immunofluorescence) |
| Antibody | Rabbit polyclonal anti-Fz | *Bastock and Strutt, 2007* | N/A | (1:300 for immunofluorescence) |
| Antibody | Rabbit polyclonal anti-GFP | Abcam | cat#ab6556 ; RRID:AB_305564 | (1:4000 for immunofluorescence) |
| Antibody | Mouse monoclonal anti-α-Tubulin DM1A | Sigma | cat#T9026 ; RRID:AB_477593 | (1:10000 for immunoblotting) |
| Antibody | Mouse monoclonal anti-Actin AC40 | Sigma | cat#A4700 ; RRID:AB_476730 | (1:5000 for immunoblotting) |
| Antibody | Mouse monoclonal anti-Fmi 74 | DSHB *Usui et al., 1999* | RRID:AB_2619583 | (1:2000 for immunoblotting) |
| Antibody | Rabbit polyclonal anti-Fz | *Bastock and Strutt, 2007* | N/A | (1:250 for immunoblotting) |
| Antibody | Rabbit polyclonal anti-GFP | Abcam | cat#ab6556; RRID:AB_305564 | (1:2000 for immunoblotting) |
| Antibody | Rat polyclonal anti-Pk | *Strutt et al., 2013* | N/A | (1:200 for immunoblotting) |

*Continued on next page*

*Continued*

| Reagent type (species) or resource | Designation | Source or reference | Identifiers | Additional information |
|---|---|---|---|---|
| Genetic reagent (*D. melanogaster*) | *stbm[6]* | **Wolff and Rubin, 1998** | FlyBase: FBal0062423 | |
| Genetic reagent (*D. melanogaster*) | *dsh[1]* | **Perrimon and Mahowald, 1987** | FlyBase: FBal0003138 | |
| Genetic reagent (*D. melanogaster*) | *dgo[380]* | **Feiguin et al., 2001** | FlyBase: FBal0141190 | |
| Genetic reagent (*D. melanogaster*) | *pk[pk-sple13]* | **Gubb et al., 1999** | FlyBase: FBal0060943 | |
| Genetic reagent (*D. melanogaster*) | *fmi[E59]* | **Usui et al., 1999** | FlyBase: FBal0101421 | |
| Genetic reagent (*D. melanogaster*) | *fz[P21]* | **Jones et al., 1996** | FlyBase:FBal0004937 | |
| Genetic reagent (*D. melanogaster*) | *P[w+, Actin5C-FRT-PolyA-FRT-fz-eYFP]* | **Strutt, 2001** | FlyBase: FBtp0017633 | |
| Genetic reagent (*D. melanogaster*) | *ActP-FRT-polyA-FRT-fz-mKate2-sfGFP* | **Ressurreição et al., 2018** | FlyBase: FBal0361851 | |
| Genetic reagent (*D. melanogaster*) | *P[w+, Act >STOP > fz-mKate2-sfGFP]* | **Ressurreição et al., 2018** | FlyBase: FBal0361851 | |
| Genetic reagent (*D. melanogaster*) | *hsFLP[attP2]* | Bloomington *Drosophila* Stock Center | FlyBase: FBti0160508 | |
| Genetic reagent (*D. melanogaster*) | *hsFLP1* | Bloomington *Drosophila* Stock Center | FlyBase: FBti0002044 | |
| Genetic reagent (*D. melanogaster*) | *hsFLP22* | Bloomington *Drosophila* Stock Center | FlyBase: FBti0000785 | |
| Genetic reagent (*D. melanogaster*) | *P[w+, hh-GAL4]* | **Tanimoto et al., 2000** | FlyBase: FBal0121962 | |
| Genetic reagent (*D. melanogaster*) | *P[w+, UAS >STOP > FzNL]* | This work (on III) | | *P[w+, UAS>STOP>FzNL]* flies can be obtained by request from the corresponding author |
| Genetic reagent (*D. melanogaster*) | *w[1118]* | Bloomington *Drosophila* Stock Center | FlyBase: FBal0018186 | |
| Genetic reagent (*D. melanogaster*) | *P[acman]-EGFP-Dgo* | **Ressurreição et al., 2018** | FlyBase: FBrf0237330 | |
| Chemical compound, drug | 16% paraformaldehyde solution (methanol free) | Agar Scientific | cat#R1026 | |
| Chemical compound, drug | Triton X-100 | VWR | cat#28817.295 ; CAS: 9002-93-1 | |
| Chemical compound, drug | Tween-20 | VWR | cat#437082Q ; CAS: 9005-64-5 | |
| Chemical compound, drug | Glycerol | VWR | cat#284546F ; CAS: 56-81-5 | |
| Chemical compound, drug | DABCO | Fluka | cat#33480 ; CAS: 280-57-9 | |
| Chemical compound, drug | Vectashield Mounting Medium H-1000 | Vector Labs | cat#H-1000-10 ; RRID:AB_2336789 | |
| Chemical compound, drug | Normal Goat Serum | Jackson Labs | cat#005-000-121 ; RRID:AB_2336990 | |
| Chemical compound, drug | Alexa568-Phalloidin | Thermo | cat#A12380 ; RRID:AB_3096418 | (1:1000) |
| Software, algorithm | ImageJ 1,54f | https://imagej.nih.gov/ij/ **Schneider et al., 2012** | | |

*Continued on next page*

*Continued*

| Reagent type (species) or resource | Designation | Source or reference | Identifiers | Additional information |
|---|---|---|---|---|
| Software, algorithm | MatLab R2019b | https://uk.mathworks.com/products/matlab.html | | |
| Software, algorithm | Puncta selection script (MatLab) | *Strutt et al., 2016* | | |
| Software, algorithm | Microtubules orientation script (MatLab) | *Ramírez-Moreno et al., 2023* | | |
| Software, algorithm | GraphPad Prism v9 | https://www.graphpad.com | | |
| Software, algorithm | QuantifyPolarity | *Tan et al., 2021* | | |
| Software, algorithm | NIS Elements AR 4.60 | Nikon | | |
| Software, algorithm | Tissue Analyzer | https://grr.gred-clermont.fr/ labmirouse/software/WebPA/ *Etournay et al., 2016* | | |

## Fly genetics and husbandry

*Drosophila melanogaster* were raised on standard cornmeal/agar/molasses media at 25°C unless otherwise specified. To over-express constructs of interest, the UAS/GAL4 system was used (*Brand and Perrimon, 1993*), with the *hedgehog-GAL4* (*hh-GAL4*) driver. Prepupae of the appropriate genotype were selected and aged at 25°C as indicated.

For induction and recombination experiments *hsFLP* was used to excise an *FRT-Stop-FRT* cassette to allow ubiquitous expression of *Fz-eYFP* (P [w+, Actin>STOP>fz-eYFP]) or *Fz-mKate2-sfGFP* (P[w+, Actin>STOP>fz-mKate2-sfGFP]attP2 fz$^{P21}$), or *hh-GAL4* controlled expression of full-length Fz protein (P[w+, UAS>STOP>FzNL]). Heat shocks were performed for 2 hr at 37°C at 24 hr APF for MT experiments, or for 1 hr at 37°C at the specified time for other experiments. On the basis of when trichomes subsequently emerge, we observe that development is halted for the period of the heat shock. Pupae were left to age at 25°C before dissecting and were fixed at the same developmental time of 28 hr APF unless otherwise stated.

Mutant alleles are described in FlyBase; *stbm*$^6$, *pk-sple*$^{13}$, *fmi*$^{E59}$, *fz*$^{P21}$, and *dgo*$^{380}$ are putative null alleles and unable to give rise to functional proteins, while *dsh*$^1$ gives rise to a mutant protein defective for planar polarity but which functions normally in Wingless signalling (*Axelrod et al., 1998*; *Boutros et al., 1998*; *Warrington et al., 2017*). Importantly, by the criterion of abolishing detectable asymmetric localisation of the other core proteins or their downstream effectors, the *stbm*$^6$, *pk-sple*$^{13}$, *fmi*$^{E59}$, *fz*$^{P21}$, and *dsh*$^1$ alleles are completely lacking in planar polarity function (*Wong and Adler, 1993*; *Usui et al., 1999*; *Axelrod, 2001*; *Strutt, 2001*; *Tree et al., 2002*; *Bastock et al., 2003*; *Strutt and Warrington, 2008*; *Lu et al., 2010*; *Warrington et al., 2017*). P[w+, UAS>STOP>FzNL] was generated by cloning the full-length *fz* coding sequence into the vector pUASt-UAS-FRT-Stop-FRT followed by P-element transformation as described in *Ramírez-Moreno et al., 2023*.

Full genotypes for each experiment are provided in *Supplementary file 1*.

## Materials availability statement

The newly created P[w+, UAS>STOP>FzNL] fly strain used in this work is freely available from the corresponding author upon request.

## Dissection and immunolabelling of pupal wings

Pupal wings were dissected at 28 hr APF at room temperature. Briefly, pupae were removed from their pupal case and fixed for 40 min in 4% paraformaldehyde in PBS, or 50 min in 10% paraformaldehyde in PBS for MT immunolabelling. Wings were then dissected and the outer cuticle removed, fixed for 10 min in 4% paraformaldehyde in PBS or in 10% paraformaldehyde in PBS for MT immunolabelling and were blocked for 1 hr in PBS containing 0.2% Triton X-100 (PTX) and 10% normal goat serum. Primary and secondary antibodies were incubated overnight at 4°C in PTX with 10% normal goat serum, and all washes were in PTX. After immunolabelling, wings were post-fixed in 4% paraformaldehyde in PBS for 10 min. Wings were mounted in 12.5 µl of Vectashield or in 25 µl of PBS containing 10% glycerol and 2.5% DABCO, pH 7.5 for MT experiments.

## Imaging

Pupal wings were oriented along the proximo-distal axis using longitudinal vein 4 as the reference and were imaged on a Nikon A1R GaAsP confocal microscope using a 60x NA1.4 apochromatic lens. Imaging of Fz::mKate2-sfGFP was done on live pupae, where a small piece of cuticle was removed from above the pupal wing, and the exposed wing was mounted on strips of double-sided tape in a glass-bottomed dish. Wings were imaged with a pixel size of 80 nm, and the pinhole was set to 1.2 AU. Nine Z-slices separated by 150 nm were imaged, and then the three brightest slices around junctions were selected and averaged for each channel in ImageJ.

For MT imaging, fixed pupal wings were imaged with a pixel size of 40 nm, and the pinhole was set to 0.4 AU. Nineteen Z-slices separated by 70 nm were captured. MT images were then deconvolved following Landweber's method with point scan confocal modality with 10 iterations. The three brightest slices around junctions were selected from the deconvolved image stack and averaged for each channel in ImageJ.

## Prehairs

For prehair visualisation, prepupa were aged for 22 hr at 27°C. Heat shocks were performed for 1 hr at 37°C. Pupa were left to age at 27°C before dissecting and were fixed at the same developmental time of 33 hr APF. Wings were processed for immunofluorescence as described for core protein detection, except to improve labelling of actin structures, wings were fixed for 50 min in supplemented fix with 1% Triton X-100; GFP protein was detected using 1:4000 rabbit anti-GFP (Abcam) and Actin was visualised using 1:1000 Alexa568-Phalloidin. Wings were oriented along the proximo-distal axis using longitudinal vein 4 as the reference and were imaged on a Nikon A1R GaAsP confocal microscope using a 60x NA1.4 apochromatic lens. Nine Z-slices separated by 150 nm were imaged, and then the three brightest slices around junctions were selected and averaged for each channel in ImageJ.

## Western blotting

For pupal wing western blots, 28 hr APF pupal wings were dissected directly into sample buffer (141 mM Tris base, 2% lithium dodecyl sulphate, 10% glycerol, 0.51 mM EDTA, 100 mM dithiothreitol, pH 8.5) and the equivalent of the indicated number of wings per lane were run. A Bio-Rad ChemiDoc XRS+ was used for imaging, and band intensities from four biological replicates were quantified using ImageJ. Data were compared using unpaired *t*-tests or ANOVA for multiple comparisons. Western blots were probed with mouse monoclonal anti-Fmi 74 (DSHB, *Usui et al., 1999*), affinity-purified rabbit anti-Fz (*Bastock and Strutt, 2007*), affinity-purified rat anti-Pk (*Strutt et al., 2013*), affinity-purified rabbit anti-GFP (Abcam ab6556), and mouse monoclonal anti-Tubulin DM1A (Sigma-Aldrich T9026). We do not have an anti-Dgo antibody suitable for western blotting.

## Adult wings

Adult wings were dehydrated in isopropanol and mounted in GMM (50% methylsalicylate, 50% Canada Balsam), and incubated overnight on a 60°C hot plate to clear. Wings were photographed under brightfield illumination on a Leica DMR compound at 20× magnification, with longitudinal vein 4 as the horizontal reference.

## Quantification and statistical analysis

### Fluorescence detection and quantitation

Membrane masks were generated using Tissue Analyzer (*Etournay et al., 2016*), and the mean intensity of fluorescence in membranes was measured using an automated MATLAB script (see **Puncta_defined_area** in MATLAB scripts in Supplemental Material and *Strutt et al., 2016*).

For *Figure 6*, cell boundary quantitations, fluorescence on medio-lateral oriented junctions or on horizontal oriented junctions was measured by manual drawing of ROIs in ImageJ. All boundaries were classified following their orientation compared to the Fz overexpression boundary generated with the UAS-GAL4 system using *hedgehog-GAL4*, with the *hedgehog* expression domain in the posterior part of the pupal wing. Horizontal junctions were defined as parallel to the Fz overexpression boundary (between 0° and 45°) and medio-lateral junctions as junctions linking two horizontal boundaries (between 45° and 90°).

For live imaging, background due to autofluorescence was subtracted, and mean fluorescence intensity was averaged across wings.

The fraction of stable Fz was determined by the ratio of mKate2/sfGFP fluorescence intensity. When all genotypes were compared, an ANOVA with Tukey's multiple comparisons test was used, or ANOVAs with Dunnett's or Kruskal–Wallis multiple comparisons tests were used to compare wild-type to mutant backgrounds.

Total Fz intensities were compared with an ANOVA with Holm–Sidak's multiple comparisons test to weigh all conditions or with unpaired *t*-test and Mann–Whitney test to compare fluorescence intensities from two conditions.

## Planar polarity

For polarity magnitude and cell morphological properties analysis, QuantifyPolarity software was used (*Tan et al., 2021*). Wing images were oriented along the proximo-distal wing axis based on longitudinal wing vein orientation corresponding to the *x* axis, and border masks were generated using Tissue Analyzer (*Aigouy et al., 2010*) to delimit cells. A cell-by-cell analysis was performed, with polarity magnitude and orientation calculated using the PCA option in QuantifyPolarity. Results are given for individual cell and average per wing. In this algorithm, a cell is first transformed onto an ellipse shape to obtain cell orientation and eccentricity. Then, for polarity measurement, intensities of individual pixels are normalised and the polarity angle is calculated as the angle that produces the largest variance of normalised intensities. Finally, the normalised intensities are converted into pseudo-*XY*-coordinates and the eigenvalues of the covariation matrix of these transformed coordinates are used to compute the polarity magnitude. Planar polarity quantification is independent of cell geometry. The full description of the algorithm is available in *Tan et al., 2021*.

For *Figure 5*, custom MATLAB scripts were used to combine polarity magnitude and orientation (see **Combined results for polarity** in MATLAB scripts in Supplemental Material), and these were visualised on polar plots (see **Polar plot for polarity** for details in MATLAB scripts in Supplemental Material). Statistical tests are described in legend text for each figure. For polarity magnitude comparisons, when all genotypes were compared between them, an ANOVA with Tukey's multiple comparisons test was used, or an ANOVA with Dunnett's multiple comparisons tests was used to compare wild-type to mutant backgrounds, or unpaired Mann–Whitney tests were used to compare with or without repolarisation induction. To compare two conditions for total Fz polarity (orientation and magnitude), Hotelling's *T*-square tests were used.

## Cell and MT orientation

Analysis of cell orientation and apical MT orientation used custom automated MATLAB image analysis scripts, based on an initial protocol developed by *Gomez et al., 2016*, *Płochocka et al., 2021*, and improved by *Ramírez-Moreno et al., 2023* (see **Microtubule_orientation** in MATLAB scripts in Supplemental Material). When all genotypes were compared between them, an ANOVA with Tukey's multiple comparisons test was used, or unpaired Mann–Whitney tests were used to compare with or without repolarisation induction.

## Description of MATLAB scripts

### Puncta_defined_area

This is a MATLAB script that measures membrane and puncta intensities on segmented images (*Strutt et al., 2016*) generated using Tissue Analyzer (*Etournay et al., 2016*), and was used here to measure total membrane intensity (Mmean).

## Microtubule_orientation

The average projections of images with tubulin signals were adjusted so that 0.5% of the pixels with the lowest intensities were set to black and the 0.5% of the pixels with the highest intensities were set to white to normalise the variability in signal between images and increase the contrast. Masks generated using Tissue Analyzer (*Etournay et al., 2016*) were used to isolate tubulin signals in individual cells. Then, the magnitude of the tubulin signal according to its direction (gradient of the signal) in each cell was calculated by convolving the tubulin signal using two 5 × 5 Sobel operators (*Gomez et al., 2016*). The resulting distributions of tubulin signals were aligned to their maxima at

0 and averaged for cells with specified eccentricities (0.750 ± 0.025 and 0.800 ± 0.025) to produce average profiles of MT angle distributions in cells (*Płochocka et al., 2021*). At the same time, the unaltered distributions were fitted with the Von Mises distribution and the estimated mean and standard deviation of the fitted curve in each cell. The mean was used as the main direction of the apical MT network in this cell and the standard deviation as the measure of the MT alignment with each other (microtubule standard deviation, MTSD). Finally, the cell directions were calculated by fitting the pixel coordinates of each cell isolated using masks to an ellipse and obtaining the direction of the long axis of the best-fit ellipse. MTSD was plotted against cell eccentricity for cells with an MTSD <90 excluding cells with unfittable distributions of tubulin signal. This MATLAB code is available at https://github.com/nbul/Cytoskeleton/tree/master/PCP-MT (*Bulgakova, 2022*). Angle visualisation and quantification data about directions of cell elongation (the direction of the long axis of the best-fit ellipse), overall directions of MT networks and Stbm polarity angle (transferred from QuantifyPolarity) were plotted on image masks produced using Tissue Analyzer (*Etournay et al., 2016*). Here, the length of plotted lines does not reflect polarity magnitudes or length of elongation but is fixed at half of the average long cell axis. Average cell orientation within each image was used to normalise the Stbm polarity angle, individual cell orientations and MT angles for each cell in polar histograms with 9 bins in 0–180° format. Data were mirrored to ease visualisation. Normalised angle data was also used to separate angles into quadrants: between –45°, 45° and between 45° and 135°. The plotting and calculations were performed within the script for the MT organisation analysis (https://github.com/nbul/Cytoskeleton/tree/master/PCP-MT).

## Combined results for polarity

This is a MATLAB script suitable for combining polarity magnitude and angle results generated by QuantifyPolarity software (see *Supplementary file 2*).

## Polar plot for polarity

This is a MATLAB script suitable for generating polar plots showing polarity magnitude and angle, with results from 'Combined results for polarity (magnitude + angle)' MatLab script (see *Supplementary file 2*).

## Acknowledgements

Larra Trinidad, Samantha Warrington, and Carl Harrison are thanked for useful discussions and the fly room staff for excellent technical support. We are grateful to Natalia Bulgakova and Miguel Ramirez-Morena for advice on microtubule angle quantification. Imaging was performed in the Wolfson Light Microscopy Facility. The work was funded by a Wellcome Trust Senior Fellowship (210630/Z/18/Z) and a Wellcome Trust Discovery Award (305132/Z/23/Z) to DS.

## Additional information

### Funding

| Funder | Grant reference number | Author |
| --- | --- | --- |
| Wellcome Trust | 10.35802/210630 | David Strutt |
| Wellcome Trust | 10.35802/305132 | David Strutt |

The funders had no role in study design, data collection, and interpretation, or the decision to submit the work for publication. For the purpose of Open Access, the authors have applied a CC BY public copyright license to any Author Accepted Manuscript version arising from this submission.

### Author contributions

Alexandre Carayon, Conceptualization, Formal analysis, Supervision, Investigation, Methodology, Writing – original draft, Writing – review and editing; Helen Strutt, Investigation, Writing – review and

editing; David Strutt, Conceptualization, Resources, Supervision, Funding acquisition, Methodology, Writing – original draft, Writing – review and editing

#### Author ORCIDs
Alexandre Carayon ⓘ https://orcid.org/0000-0003-4440-7112
Helen Strutt ⓘ https://orcid.org/0000-0003-4365-2271
David Strutt ⓘ https://orcid.org/0000-0001-8185-4515

Reviewer #1 (Public review): https://doi.org/10.7554/eLife.107947.2.sa1
Reviewer #2 (Public Review): https://doi.org/10.7554/eLife.107947.2.sa2
Reviewer #3 (Public Review): https://doi.org/10.7554/eLife.107947.2.sa3
Author response https://doi.org/10.7554/eLife.107947.2.sa4

## Additional files

### Supplementary files
Supplementary file 1. Genotypes used in each experiment. Excel file listing genotypes of *Drosophila* strains used for each experiment.

Supplementary file 2. Matlab scripts. PDF file containing new Matlab scripts used for data analysis.

MDAR checklist

### Data availability
All source data is available as source data files within the manuscript.

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
