## [Editor Report · eLife Assessment]

This **useful** paper examined the mechanism of planar cell polarity (PCP) using *Drosophila* pupal wing, investigating how 'cellular level', 'molecular level' and 'tissue level' mechanisms intersect to establish PCP. This represents progress for the field and the conclusions are mostly backed up by **solid** data. Whereas the manuscript is sound overall, remaining concerns could be addressed by textual clarification of the concepts used in the manuscript.

[Editors' note: this paper was reviewed by Review Commons and revised by the authors.]

---

## [Referee Report · Reviewer #1 (Public review)]

The authors use inducible Fz::mKate2-sfGFP to explore "cell-scale signaling" in PCP. They reach several conclusions. First, they conclude that cell-scale signaling does not depend on limiting pools of core components (other than Fz). Second, they conclude that cell-scale signaling does not depend on microtubule orientation, and third, they conclude that cell-scale signaling is strong relative to cell to cell coupling of polarity.

There are some interesting inferences that can be drawn from the manuscript, but there are also some significant challenges in interpreting the results and conclusions from the work as presented. I suggest that the authors (1) define "cell-scale signaling," as the precise meaning must be inferred, (2) reconsider some premises upon which some conclusions depend, (3) perform an essential assay validation, and (4) explain some other puzzling inconsistencies.

Major concerns (first round of review):

The exact meaning of cell-scale signaling is not defined, but I infer that the authors use this term to describe how what happens on one side of a cell affects another side. The remainder of my critique depends on this understanding of the intended meaning.

The authors state that any tissue wide directional information comes from pre-existing polarity and its modification by cell flow, such that the de novo signaling paradigm "bypasses" these events and should therefore not be responsive to any further global cues. It is my understanding that this is not a universally accepted model, and indeed, the authors' data seem to suggest otherwise. For example, the image in Fig 5B shows that de novo induction restores polarity orientation to a predominantly proximal to distal orientation. If no global cue is active, how is this orientation explained? The 6 hr condition, that has only partial polarity magnitude, is quite disordered. Do the patterns at 8 and 10 hrs become more proximally-distally oriented? It is stated that they all show swirls, but please provide adult wing images, and the corresponding orientation outputs from QuantifyPolarity to help validate the notion that the global cues are indeed bypassed by this paradigm.

It is implicit that, in the de novo paradigm, polarization is initiated immediately or shortly after heat shock induction. However, the results should be differently interpreted if the level of available Fz protein does not rise rapidly and then stabilize before the 6 hr time point, and instead continues to rise throughout the experiment. Western blots of the Fz::mKate2-sfGFP at time points after induction should be performed to demonstrate steady state prior to measurements. Otherwise, polarity magnitude could simply reflect the total available pool of Fz at different times after induction. Interpreting stability is complex, and could depend on the same issue, as well as the amount of recycling that may occur. Prior work from this lab using FRAP suggested that turnover occurs, and could result from recycling as well as replenishment from newly synthesized protein.

From the Fig 3 results, the authors claim that limiting pools of core proteins do not explain cell-scale signaling, a result expected based on the lack of phenotypes in heterozygotes, but of course they do not test the possibility that Fz is limiting. They do note that some other contributing protein could be.

In Fig 3, it is unclear why the authors chose to test dsh1/+ rather than dsh[null]/+. In any case, the statistically significant effect of Dsh dose reduction is puzzling, and might indicate that the other interpretation is correct. Ideally, a range including larger and smaller reductions would be tested. As is, I don't think limiting Dsh is ruled out.

The data in Fig 5 are somewhat internally inconsistent, and inconsistent with the authors' interpretation. In both repolarization conditions, the authors claim that repolarization extends only to row 1, and row 1 is statistically different from non-repolarized row 1, but so too is row 3. Row 2 is not. This makes no sense, and suggests either that the statistical tests are inappropriate and/or the data is too sparse to be meaningful. For the related boundary intensity data in Fig 6, the authors need to describe exactly how boundaries were chosen or excluded from the analysis. Ideally, all boundaries would be classified as either meido-lateral (meaning anterior-posterior) or proximal-distal depending on angle.

If the authors believe their Fig 5 and 6 analyses, how do they explain that hairs are reoriented well beyond where the core proteins are not? This would be a dramatic finding, because as far as I know, when core proteins are polarized, prehair orientation always follows the core protein distribution. Surprisingly, the authors do not so much as comment about this. The authors should age their wings just a bit more to see whether the prehair pattern looks more like the adult hair pattern or like that predicted by their protein orientation results.

---

## [Referee Report · Reviewer #2 (Public Review)]

This paper aims to dissect the relative importance of the various cues that establish PCP in the wing disc of Drosophila, which remains a prominent and relevant model for PCP. The authors suggest that one must consider cues at three scales (molecular, cell and tissue) and specifically design tests for the importance of cell-level cues, which they call non-local cell scale signalling. They develop clever experimental approaches that allow them to track complex stability and also to induce polarity at experimentally defined times. In a first set of experiments, they restore PCP after the global cues have disappeared (de novo polarisation) and conclude from the results that another (cell scale) cue must exist. In another set of experiments, they show that de novo repolarization is robust to the dosage of various components of core PCP, leading them to conclude that there must be an underlying cell scale polarity, which, apparently, has nothing to do with microtubule or cell shape polarity. They then describe nice evidence that de novo polarisation is relatively short range both in a polarised and unpolarised field. They conclude that there is a strong cell-intrinsic polarity that remains to be characterised.

Major concerns (first round of review):

(1) The first set of repolarisation experiments is performed after the global cell rearrangements that have been shown to act as global signals. However, this approach does not exclude the possible contribution of an unknown diffusible global signal.

(2) The putative non-local cell scale signal must be more precisely defined (maybe also given a better name). It is not clear to me that one can separate cell-scale from molecular-scale signal. Local signals can redistribute within a cell (or membrane) so local signals are also cell-scale. Without a clear definition, it is difficult to interpret the results of the gene dosage experiments. The link between gene dosage and cell-scale signal is not rigorously stated. Related to this, the concluding statement of the introduction is too cryptic.

Critique:

The experiments described in this paper are of high quality with a sophisticated level of design and analysis. However, there needs to be some recalibration of the extent of the conclusions that can be drawn. Moreover, a limitation of this paper is that, despite the quality of their data, they cannot give a molecular hint about the nature of their proposed cell-scale signal.

---

## [Referee Report · Reviewer #3 (Public Review)]

The manuscript by Carayon and Strutt addresses the role of cell-scale signaling during the establishment of planar cell polarity (PCP) in the Drosophila pupal wing. The authors induce locally the expression of a tagged core PCP protein, Frizzled, and observe and analyze the de novo establishment of planar cell polarity. Using this system, the authors show that PCP can be established within several hours, that PCP is robust towards variation in core PCP protein levels, that PCP proteins do not orient microtubules, and that PCP is robust towards 'extrinsic' re-polarization. The authors conclude that the polarization at the cell-scale is strongly intrinsic and only weakly affected by the polarity of neighboring cells.

Major comments (first round of review):

The data are clearly presented and the manuscript is well written. The conclusions are well supported by the data.

(1) The authors use a system to de novo establish PCP, which has the advantage of excluding global cues orienting PCP and thus to focus on the cell-intrinsic mechanisms. At the same time, the system has the limitation that it is unclear to what extent de novo PCP establishment reflects 'normal' cell scale PCP establishment, in particular because the Gal4/UAS expression system that is used to induce Fz expression will likely result in much higher Fz levels compared with the endogenous levels. The authors should briefly discuss this limitation.

(2) Fig. 3. The authors use heterozygous mutant backgrounds to test the robustness of de novo PCP establishment towards (partial) depletion in core PCP proteins. The authors conclude that de novo polarization is 'extremely robust to variation in protein level'. Since the authors (presumably) lowered protein levels by 50%, this conclusion appears to be somewhat overstated. The authors should tune down their conclusion.

Significance:

The manuscript contributes to our understanding of how planar cell polarity is established. It extends previous work by the authors (Strutt and Strutt, 2002,2007) that already showed that induction of core PCP pathway activity by itself is sufficient to induce de novo PCP. This manuscript further explores the underlying mechanisms. The authors test whether de novo PCP establishment depends on an 'inhibitory signal', as previously postulated (Meinhardt, 2007), but do not find evidence. They also test whether core PCP proteins help to orient microtubules (which could enhance cell intrinsic polarization of core PCP proteins), but, again, do not find evidence, corroborating previous work (Harumoto et al, 2010). The most significant finding of this manuscript, perhaps, is the observation that local de novo PCP establishment does not propagate far through the tissue. A limitation of the study is that the mechanisms establishing intrinsic cell scale polarity remain unknown. The work will likely be of interest to specialists in the field of PCP.

**Summary of comments from the Reviewing Editor on the revised version:**

In the introduction, when you refer to Figure 1, the definition of Molecular, cellular, tissue scale is indeed not too clear to outside readers. For example, when you first refer to 'cell scale' you define it 'non-local', but probably it is not clear to many readers 'non-local' means 'the mechanism that cannot be explained by 'molecular scale'. (because 'molecular scale = local' is only inferred).

The 'conclusion paragraph' at the end of the Introduction does not have conclusion (only explained 'which question was tested by which method').

Minor comments that can easily be addressed by textual edits:

– they do not explain why gene dosage affects constitutive but not de novo polarization. It seems to me that one would expect de novo to be at least as sensitive if not more.

– Unconventional nomenclature for tissue axes - mediolateral, horizontal - are frequently used. These are sometimes difficult to parse. Please stick with universally accepted anterior, posterior, proximal and distal.

---

## [Author Response]

**(1) General Statements**

Our manuscript studies mechanisms of planar polarity establishment in vivo in the *Drosophila* pupal wing. Specifically we seek to understand mechanisms of ‘cell-scale signalling’ that is responsible for segregating core pathway planar polarity proteins to opposite cell edges. This is an understudied question, in part because it is difficult to address experimentally.

We use conditional and restrictive expression tools to spatiotemporally manipulate core protein activity, combined with quantitative measurement of core protein distribution, polarity and stability. Our results provide evidence for a robust cell-scale signal, while arguing against mechanisms that depend on depletion of a limited pool of a core protein or polarised transport of core proteins on microtubules. Furthermore, we show that polarity propagation across a tissue is hard, highlighting the strong intrinsic capacity of individual cells to establish and maintain planar polarity.

The original manuscript received three fair and thorough peer-reviews, which raised many important points. In response, we decided to embark on a full revision that attempts to answer all of the points. We have included new data to support our conclusions in Supplemental Figures 1, 2 and 5.

Additionally in response to the reviewers we have revised the manuscript title, which is now ‘Characterisation of cell-scale signalling by the core planar polarity pathway during *Drosophila* wing development’.

**(2) Point-by-point description of the revisions**

We thank all of the reviewers for their thorough and thoughtful review of our manuscript. They raise many helpful points which have been extremely useful in assisting us to revise the manuscript.

In response we have carried out a major revision of the manuscript, making numerous changes and additions to the text and also adding new experimental data. Specific changes are listed after our detailed response to each comment.

**Reviewer #1:**
[…] Major points:The exact meaning of cell-scale signaling is not defined, but I infer that the authors use this term to describe how what happens on one side of a cell affects another side. The remainder of my critique depends on this understanding of the intended meaning.

As the reviewer points out, it is important that the meaning of the term ‘cell-scale signalling’ is clear to the reader and in response to their comment we have had another go at defining it explicitly in the Introduction to the manuscript.

Specifically, we use the term ‘cell-scale signalling’ to describe possible intracellular mechanisms acting on core protein segregation to opposite cell membranes during core pathway dependent planar polarisation. For example, this could be a signal from distal complexes at one side of the cell leading to segregation of proximal complexes to the opposite cell edge, or vice versa. See also our response to Reviewer #2 regarding the distinction between ‘molecular-scale’ and ‘cell-scale’ signalling.

Changes to manuscript: Revised definition of ‘cell-scale signalling’ in Introduction.

The authors state that any tissue wide directional information comes from pre-existing polarity and its modification by cell flow, such that the de novo signaling paradigm "bypasses" these events and should therefore not be responsive to any further global cues. It is my understanding that this is not a universally accepted model, and indeed, the authors' data seem to suggest otherwise. For example, the image in Fig 5B shows that de novo induction restores polarity orientation to a predominantly proximal to distal orientation. If no global cue is active, how is this orientation explained?

We assume that the reviewer’s point is that it is not universally accepted that de novo induction after hinge contraction leads to uncoupling from global cues (rather than that it is not accepted that hinge contraction remodels radial polarity to a proximodistal pattern). We are (we believe) the only lab that has used *de novo* induction as a tool, and we’re not aware of any debate in the literature about whether this bypasses global cues. Nevertheless, we accept that it is hard to prove there is *no influence* of global cues, when the nature of those cues and the time at which they act remain unclear. Below we summarise the reasons why we believe there are not significance effects of global cues in our experiments that would influence the interpretation of our results.

First, our reading of the literature supports a broad consensus that an early radial core planar polarity pattern is realigned by cell flow produced by hinge contraction beginning at around 16h APF (e.g. Aigouy et al., 2010; Strutt and Strutt, 2015; Aw and Devenport, 2017; Butler and Wallingford, 2017; Tan and Strutt, 2025). Taken at face value, this suggests that there are ‘radial’ cues present prior to hinge contraction, maybe coming from the wing margin – arguably these radial cues could be Ft-Ds or Wnts or both, given they are expressed in patterns consistent with such a role (notwithstanding the published evidence arguing against roles for either of these cues). It then appears that hinge contraction supercedes these cues to convert a radial pattern to a proximodistal pattern – whether the radial cues that affect the core pathway earlier remain active after hinge contraction is unclear, although both Ft-Ds and Wnts appear to maintain their ‘radial’ patterns beyond the beginning of hinge contraction (e.g. Merkel et al., 2014; Ewen-Campen et al., 2020; Yu et al., 2020).

We think that the reviewer is proposing the presence of a proximodistal cue that is active in the proximal region of the wing that we use for our experiments shown e.g. in Fig.5, and that this cue orients core polarity here (but not elsewhere in the wing) in a time window after 18h APF. Ft-Ds and Wnts do not seem to be plausible candidates as they are still in ‘radial’ patterns. This leaves either an unknown proximodistal cue (a gradient of some unknown signalling molecule?), or possibly some ability of hinge contraction to align proximodistal polarity specifically in this wing region but not elsewhere. We cannot definitively rule out either of these possibilities, but neither do we think there is sufficient evidence to justify invoking their existence to explain our observations.

In particular, the reason that we don’t think there is a proximodistal cue in the proximal part of the wing after 18h APF, is that work from our lab shows that induction of Fz or Stbm expression at times around or after the start of hinge contraction (i.e. >16 h APF) results in increasing levels of trichome swirling with polarity not being coordinated with the tissue axis either proximally or distally (Strutt and Strutt, 2002; Strutt and Strutt 2007). Our simplest interpretation for this is that induction at these stages fails to establish the early radial pattern of core pathway polarity and hence hinge contraction cannot reorient radial to proximodistal. If hinge contraction alone could specify proximodistal polarity in the absence of the earlier radial polarity, then we would not expect to see swirling over much of the proximal wing (where the forces from hinge contraction are strongest (Etournay et al., 2015)).

In this manuscript, our earliest *de novo* experiments begin with Fz induction at 18h APF (*de novo* 10h), then at 20h APF (*de novo* 8h) and at 22h APF (*de novo* 6h). The image in Fig. 5B, referred to by the reviewer, is of a wing where Fz is induced *de novo* at 22 h APF. In these wings, as expected, the core proteins localise asymmetrically in stereotypical swirling patterns throughout the wing surface (see Fig. 2M and also Strutt and Strutt, 2002; Strutt and Strutt 2007), but – usefully for our experiments – they broadly localise along the proximal-distal axis in the region analysed in Fig. 5B. Given the strong swirling in surrounding regions when inducing at >20h APF, we feel reasonably confident in assuming that the pattern is not due to a proximodistal cue present in the proximal wing.

We appreciate that the original manuscript did not show images including the trichome pattern in adjacent regions, so this point would not have been clear, but we now include these in Supplementary Fig. 5. We have also added a note in the legend to Fig. 5B to clarify that the proximodistal pattern seen is local to this wing region. We apologise for this oversight and the confusion caused and appreciate the feedback.

The 6 hr condition, that has only partial polarity magnitude, is quite disordered. Do the patterns at 8 and 10 hrs become more proximally-distally oriented? It is stated that they all show swirls, but please provide adult wing images, and the corresponding orientation outputs from QuantifyPolarity to help validate the notion that the global cues are indeed bypassed by this paradigm.

In all three ‘normal’ *de novo* conditions (6h, 8h and 10h), regardless of the time of induction, the polarity orientation patterns of Fz-mKate2 in pupal and adult wings are very similar in the experimentally analysed region (Fig. S5B-E). The strong local hair swirling agrees with the previous published data (Strutt and Strutt, 2002; Strutt and Strutt 2007). Overall, we don’t see any evidence that the 10h *de novo* induction results in more proximodistally coordinated polarity than the 8h or 6h conditions. This is consistent with our contention that there is no global cue present at these stages, which presumably would have a stronger effect when core pathway activity was induced at earlier stages.

Changes to manuscript: Added additional explanation of the ‘de novo induction’ paradigm and why we believe the resulting polarity patterns are unlikely to be influenced by any global signals in Introduction and Results section ‘Induced core protein relocalisation…’. Added quantification of polarity in the experiment region proximal to the anterior cross-vein in pupal wings (Fig.S5E-E’’’) and zoomed-out images of the surrounding region in adult wings showing that the polarity pattern does not become more proximodistal when induction time is longer, and also that there is not overall proximodistal polarity in proximal regions of the wing (Fig.S5B-D), arguing against an unknown proximodistal polarity cue at these stages of development.

In the de novo paradigm, polarization is initiated immediately or shortly after heat shock induction. However, the results should be differently interpreted if the level of available Fz protein does not rise rapidly and then stabilize before the 6 hr time point, and instead continues to rise throughout the experiment. Western blots of the Fz::mKate2-sfGFP at time points after induction should be performed to demonstrate steady state prior to measurements. Otherwise, polarity magnitude could simply reflect the total available pool of Fz at different times after induction. Interpreting stability is complex, and could depend on the same issue, as well as the amount of recycling that may occur. Prior work from this lab using FRAP suggested that turnover occurs, and could result from recycling as well as replenishment from newly synthesized protein.

The reviewer raises an important point, which we agree could confound our experimental interpretations. As suggested we have now carried out western blotting and quantitation for Fz::mKate2-sfGFP levels and added these data to Fig.S1 (Fig. S1C,D). Quantified Fz is not significantly different between the three *de novo* polarity induction timings and not significantly different compared to constitutive Fz::mKate2-sfGFP expression (although there is a trend towards increasing Fz::mKate2-sfGFP protein levels with increasing induction times). These data are consistent with Fz::mKate2-sfGFP being at steady state in our experiments and that levels are sufficient to achieve normal polarity (as constitutive Fz::mKate2-sfGFP does so). Therefore it is unlikely that differing protein levels explain the differing polarity magnitudes at the different induction times. Interestingly, Fz::mKate2-sfGFP levels are lower than endogenous Fz levels, possibly due to lower expression or increased turnover/reduced recycling.

Changes to manuscript: Added western blot analysis of Fz::mKate2-sfGFP expression under 10h, 8h and 6h induction conditions vs endogenous Fz expression and constitutive Fz::mKate2sfGFP expression (Fig.S1C-D) and discussed in Results section ‘Planar polarity establishment is…’.

From the Fig 3 results, the authors claim that limiting pools of core proteins do not explain cellscale signaling, a result expected based on the lack of phenotypes in heterozygotes, but of course they do not test the possibility that Fz is limiting. They do note that some other contributing protein could be.

Previously published results from our lab (Strutt et al., 2016 Cell Reports; Supplemental Fig. S6E) show that in a heterozygous *fz* mutant background, Fz protein levels are not affected by halving the gene dosage when compared to wt, suggesting that Fz is most likely produced in excess and is not normally limiting, but that protein that cannot form complexes may be rapidly degraded. We have now added this information to the text.

Changes to manuscript: Added explanation in text that Fz levels had previously been shown to not be dosage sensitive in Results section ‘Planar polarity establishment is…’ and also added a caveat to the Discussion about not directly testing Fz.

In Fig 3, it is unclear why the authors chose to test dsh1/+ rather than dsh[null]/+. In any case, the statistically significant effect of Dsh dose reduction is puzzling, and might indicate that the other interpretation is correct. Ideally, a range including larger and smaller reductions would be tested. As is, I don't think limiting Dsh is ruled out.

Concerning the choice of *dsh* allele, we appreciate the query of the reviewer regarding use of *dsh[1]* instead of a null, as there might be a concern that *dsh[1]* would give a less strong phenotype. The answer is that over more than two decades we and others have never found any evidence that *dsh[1]* does not act as a ‘null’ for planar polarity in the pupal wing, and furthermore use of *dsh[1]* preserves function in Wg signalling – and we would prefer to rule out any phenotypic effects due to any potential cross-talk between the two pathways that might be seen using a complete null. To expand on this point, *dsh[1]* mutant protein is never seen at cell junctions (Axelrod 2001; Shimada et al., 2001; our own work), and by every criteria we have used, planar polarity is completely disrupted in hemizygous or homozygous mutants e.g. see quantifications of polarity in (Warrington et al., 2017 Curr Biol).

In terms of the broader point, whether we can rule out Dsh being limiting, we were very careful to be clear that we did not see evidence for Dsh (or other core proteins) being limiting in terms of ‘rates of core pathway *de novo* polarisation’. When the reviewer says ‘the statistically significant effect of Dsh dose reduction is puzzling’ we believe they are referring to the data in Fig. 3J, showing a small but significantly different reduction in stable Fz in *de novo* 6h conditions (also seen in 8h *de novo* conditions, Fig. S3I). As Dsh is known to stabilise Fz in complexes (Strutt et al., 2011 Dev Cell; Warrington et al., 2017 Curr Biol), in itself this result is not wholly surprising. Nevertheless, while this shows that halving Dsh levels does modestly reduce Fz stability, it does not alter our conclusion that halving Dsh levels does not affect Fz polarisation rate under either 6h or 8h *de novo* conditions.

Unfortunately, we do not have available to us a practical way of achieving consistent intermediate reductions in Dsh levels (e.g. a series of verified transgenes expressing at different levels). Levels of all the core proteins could be dialled down using transgenes, to see when the system breaks, and indeed we have previously published that lower levels of polarity are seen if Fmi levels are <<50% or if animals are transheterozygous for *pk, stbm, dgo* or *dsh, pk, stbm, dgo* simultaneously (Strutt et al., 2016 Cell Reports). However, it seems to be a trivial result that eventually the ability to polarise is lost if insufficient core proteins are present at the junctions. For this reason we have focused on a simple set of experiments reducing gene dosage singly by 50% under two *de novo* induction conditions, and have been careful to state our results cautiously. The assays we carried out were a great deal of work even for just the 5 heterozygous conditions tested.

We believe that the experiments shown effectively make the point that there is no strong dosage sensitivity – and it remains our contention that if protein levels were the key to setting up cell-scale polarity, then a 50% reduction would be expected to show an effect on the rate of polarisation. We further note that as Fz::mKate2-sfGFP levels are lower than endogenous Fz levels (see above), the system might be expected to be sensitised to further dosage reductions, and despite this we failed to see an effect on rate of polarisation.

We note that Reviewer #3 made a similar point about whether we can rule out dosage sensitivity on the basis of 50% reductions in protein level. To address the comments of both reviewers we had now added some further narrative and caveats in the text.

In a similar vein, Reviewer #2 requested data on whether dosage reduction altered protein levels by the expected amount. We have now added further explanation/references and western blot data to address this.

Changes to manuscript: Added more explanation of our choice of *dsh[1]* as an appropriate mutant allele to use in Results section ‘Planar polarity establishment is…’. Added some narrative and caveats regarding whether lowering levels more than 50% would add to our findings in the Discussion. Revised conclusions to be more cautious including altering section title to read ‘Planar polarity establishment is not highly sensitive to variation in protein levels of core complex components’.

Also added westerns and text/references showing that for the tested proteins there is a reduction in protein levels upon removal of one gene dosage in Results section ‘Planar polarity establishment is…’ and Fig.S2.

The data in Fig 5 are somewhat internally inconsistent, and inconsistent with the authors' interpretation. In both repolarization conditions, the authors claim that repolarization extends only to row 1, and row 1 is statistically different from non-repolarized row 1, but so too is row 3. Row 2 is not. This makes no sense, and suggests either that the statistical tests are inappropriate and/or the data is too sparse to be meaningful.

As we’re sure the reviewer appreciates, this was an extremely complex experiment to perform and analyse. We spent a lot of time trying to find the best way to illustrate the results (finally settling on a 2D vector representation of polarity) and how to show the paired statistical comparisons between different groups. Moreover, in the end we were only able to detect generally quite modest (statistically significant) changes in cell polarity under the experimental conditions.

However, we note that failure to see large and consistent changes in polarity is *exactly the expected result* if it is hard to repolarise from a boundary – and this is of course the conclusion that we draw. Conversely, if repolarisation were easy, which was our expectation at least under *de novo* conditions without existing polarity, then we would have expected large and highly statistically significant changes in polarity across multiple cell rows. Hence we stand by our conclusion that ‘it is hard to repolarise from a boundary of Fz overexpression in both control and *de novo* polarity conditions’.

Overall, we were trying to establish three points:

(1) to demonstrate that repolarisation occurs from a boundary of overexpression i.e. from boundary 0 to row 0

(2) to establish whether a wave of repolarisation occurs across rows 1, 2 and 3

(3) to determine if in repolarisation in *de novo* condition it is easier to repolarise than in repolarisation in the control (already polarised) condition Taking each in turn:

(1) To detect repolarisation from a boundary relative to the control condition, we have to compare row 0 in repolarisation condition (Fig.5G,K) vs control condition (Fig.5F,J). This comparison shows a significative repolarisation (p=0.0014). From now, row 0 in repolarisation condition is our reference for repolarisation occurring.

(2) To determine if there is a wave of repolarisation in the repolarisation condition we have to compare row 0 vs row 1 to 3 in the repolarisation condition (Fig.5K). Row 1 is not significantly different to row 0, but rows 2 and 3 are different and the vectors show obviously lower polarity than row 0. Hence no wave of repolarisation is detected over rows 1 to 3.

(3) To determine if it is easier to repolarise in the *de novo condition*, our reference for establishment of a repolarisation pattern is the polarisation condition in rows 0 to 3. So, we compare repolarisation condition vs repolarisation in *de novo* condition, row 0 vs row 0, row 1 vs row 1, row 2 vs row 2 and row 3 vs row 3 – in each case no significative difference in polarity is detected, supporting our conclusion that it is not easier to repolarise in the *de novo* condition.

We agree that the variations in row 3 are puzzling, but there is no evidence that this is due to propagation of polarity from row 0, and so in terms of our three questions, it does not alter our conclusions.

Changes to manuscript: We have extensively revised the text describing the results in Fig.5 to hopefully make the reasons for our conclusions clearer and also be more cautious in our conclusions in Results section ‘Induced core protein relocalisation…’.

For the related boundary intensity data in Fig 6, the authors need to describe exactly how boundaries were chosen or excluded from the analysis. Ideally, all boundaries would be classified as either meido-lateral (meaning anterior-posterior) or proximal-distal depending on angle.

We thank the reviewer for pointing out that this was not clear.

All boundaries were classified following their orientation compared to the Fz over-expression boundary using *hh-GAL4* expressed in the wing posterior compartment. Horizontal junctions were defined as parallel to the Fz over-expression boundary (between 0 and 45 degrees) and mediolateral junctions as junctions linking two horizontal boundaries (between 45 and 90 degrees).

Changes to manuscript: The boundary classification detailed above has been added in the Materials and Methods.

If the authors believe their Fig 5 and 6 analyses, how do they explain that hairs are reoriented well beyond where the core proteins are not? This would be a dramatic finding, because as far as I know, when core proteins are polarized, prehair orientation always follows the core protein distribution. Surprisingly, the authors do not so much as comment about this. The authors should age their wings just a bit more to see whether the prehair pattern looks more like the adult hair pattern or like that predicted by their protein orientation results.

Again the reviewer makes an interesting point, and we agree that this is something that we should have more directly addressed in the manuscript.

There are three reasons why we might expect adult trichomes to show a different effect from the measured core protein polarity pattern seen in our experiments:

(i) we are assaying core protein polarity at 28h APF, but trichomes emerge at >32h APF, so there is still time for polarity to propagate a bit further from the boundary. We now have added data showing that by the point of trichome initiation, the wave of polarisation extends 3-4 cell rows (Fig.S5A).

(ii) it has long been known that a strong localisation of core proteins at a cell edge is not required for polarisation of trichome polarity from a boundary. For instance, in Strutt & Strutt 2007 we show clones of cells overexpressing Fz causing propagation through pk[pk-sple] mutant tissue where there is no detectable core protein polarity. We were following up prior observations of Adler et al., 2000 in the wing and Lawrence et al., 2004 in the abdomen.

(iii) there is evidence to suggest that the polarity of adult trichomes is locally coupled, possibly mechanically. This point is hard to prove without live imaging taking in both initial core protein localisation, the site of actin-rich trichome initiation and then the final orientation of the much larger microtubule filled trichome, and we’re not aware that such data exist. However, Wong & Adler 1993 (JCB) showed that over a number of hours trichomes become much larger and move towards the centre of the cell, presumably becoming decoupled from any core protein cue. The images in Guild … & Tilney, 2005 (MBoC) are also interesting to look at in this regard. Finally, septate junction proteins have been implicated in local alignment of trichomes, independently of the core pathway (Venema … & Auld, 2004 Dev Biol).

Changes to manuscript: Added new data in Fig.S5A showing where trichomes initiate under 6h *de novo* induction conditions, for comparison to core protein localisation and adult trichome data in Fig.5. Added some text explaining why adult trichome repolarisation might be stronger than the observed effects on core protein localisation in Discussion.

Minor points:As the authors know, there is a model in the literature that suggests microtubule trafficking provides a global cue to orient PCP. The authors' repolarization data in Fig 4 make a reasonably convincing case against a role for no role for microtubules in cell-scale signaling, but do not rule out a role as a global cue. The authors should be careful of language such as "...MTs and core proteins being oriented independently of each other" that would appear to possibly also refer to a role as a global cue.

Thank you for pointing out that this was not clear. We have now modified the text to hopefully address this.

Changes to manuscript: Text updated in Results section ‘Microtubules do not provide…’.

Significance:There are two negative conclusions and one positive conclusion made by the authors. Provided the above points are addressed, the negative conclusions, that core proteins are not limiting and that microtubules are not involved in cell-scale signaling are solid. The positive conclusion is more nebulous - the authors say that cell-scale signaling is strong relative to cell-cell signaling - but how strong is strong? Strong relative to their prior expectations? I'm not sure how to interpret such a conclusion. Overall, we learn something from these results, though it fails to reveal anything about mechanism. These results will be of some interest to those studying PCP.

The reviewer raises an interesting point, which is how do you compare the strength of two different processes, even if both processes affect the same outcome (in this case cell polarity). Repolarisation from a boundary has not been carefully studied at the level of core protein localisation in any previous study to our knowledge – this is one of the important novel aspects of this study. Hence there is not a baseline for defining strong repolarisation. Similarly, there has been no investigation of the nature of ‘cell-scale signalling’. This was a considerable challenge for us in writing the manuscript, and we have done our best to find appropriate language that hopefully conveys our message adequately. Minimally our work may provide a baseline for helping to define the ‘strengths’ of these processes in future studies.

One of our main points is that we can generate an artificial boundary of Fz expression, where Fz levels are at least several fold higher than in the neighbouring cell (e.g. compare Fig.4N’ and O’) and only two rows of cells show a significant change in polarity relative to controls. Even when the tissue next to the overexpression domain is still in the process of generating polarity (*de novo* condition) then the boundary has little effect on polarity in neighbouring cell rows. This was a result that surprised us, and we tried to convey that by using language to suggest cell-scale signalling was stronger than cell-cell signalling i.e. stronger in terms of the ability to define the final direction of polarity.

Changes to manuscript: In the revised manuscript we have reviewed our use of language and now avoid saying ‘strong’ but instead use terms such as ‘effective’ and ‘robust’ in e.g. Results section ‘Induced core protein relocalisation…’, the Discussion and we have also changed the title of the manuscript to avoid claiming a ‘strong’ signal.

**Reviewer #2:**
[…] CritiqueThe experiments described in this paper are of high quality with a sophisticated level of design and analysis. However, there needs to be some recalibration of the extent of the conclusions that can be drawn (see below). Moreover, a limitation of this paper is that, despite the quality of their data, they cannot give a molecular hint about the nature of their proposed cell-scale signal. Below are a two key points that the authors may want to clarify.(1) The first set of repolarisation experiment is performed after the global cell rearrangements that have been shown to act as global signal. However, this approach does not exclude the possible contribution of an unknown diffusible global signal.

A similar point was raised by Reviewer 1. For the convenience of this reviewer, we’ll summarise the arguments against such an unknown cue again below. More broadly, both reviewers asking a similar question indicates that we have failed to lay out the evidence in sufficient detail. In our defence, we have used the same ‘*de novo*’ paradigm in three previous publications (Strutt and Strutt 2002, 2007; Brittle et al 2022) without attracting (overt) controversy. We have now added text to the Introduction and Results that goes into more detail, as well as more experimental evidence (Fig.S5).

Firstly, it is worth noting that the global cues acting in the wing are poorly understood, with mostly negative evidence against particular cues accruing in recent years. This makes it a hard subject to succinctly discuss. Secondly, we accept that it is hard to prove there is no influence of global cues, when the nature of those cues and the time at which they act remain unclear. Below we summarise the reasons why we believe there are not significance effects of global cues in our experiments that would influence the interpretation of our results.

First, our reading of the literature supports a broad consensus that an early radial core planar polarity pattern is realigned by cell flow produced by hinge contraction beginning at around 16h APF (e.g. Aigouy et al., 2010; Strutt and Strutt, 2015; Aw and Devenport, 2017; Butler and Wallingford, 2017; Tan and Strutt, 2025). Taken at face value, this suggests that there are ‘radial’ cues present prior to hinge contraction, maybe coming from the wing margin – arguably these radial cues could be Ft-Ds or Wnts or both, given they are expressed in patterns consistent with such a role (notwithstanding the published evidence arguing against roles for either of these cues). It then appears that hinge contraction supercedes these cues to convert a radial pattern to a proximodistal pattern – whether the radial cues that affect the core pathway earlier remain active after hinge contraction is unclear, although both Ft-Ds and Wnts appear to maintain their ‘radial’ patterns beyond the beginning of hinge contraction (e.g. Merkel et al., 2014; Ewen-Campen et al.,2020; Yu et al., 2020).

We think that the reviewers are proposing the presence of a proximodistal cue that is active in the proximal region of the wing that we use for our experiments shown e.g. in Fig.5, and that this cue orients core polarity here (but not elsewhere in the wing) in a time window after 18h APF. Ft-Ds and Wnts do not seem to be plausible candidates as they are still in ‘radial’ patterns. This leaves either an unknown proximodistal cue (a gradient of some unknown signalling molecule?), or possibly some ability of hinge contraction to align proximodistal polarity specifically in this wing region but not elsewhere. We cannot definitively rule out either of these possibilities, but neither do we think there is sufficient evidence to justify invoking their existence to explain our observations.

In particular, the reason that we don’t think there is a proximodistal cue in the proximal part of the wing after 18h APF, is that work from our lab shows that induction of Fz or Stbm expression at times around or after the start of hinge contraction (i.e. >16 h APF) results in increasing levels of trichome swirling with polarity not being coordinated with the tissue axis either proximally or distally (Strutt and Strutt, 2002; Strutt and Strutt 2007). Our simplest interpretation of this is that induction at these stages fails to result in the early radial pattern of core pathway polarity being established and hence a failure of hinge contraction to reorient radial to proximodistal. If hinge contraction alone could specify proximodistal polarity in the absence of the earlier radial polarity, then we would not expect to see swirling over much of the proximal wing (where the forces from hinge contraction are strongest, Etournay et al., 2015).

In this manuscript, our earliest *de novo* experiments begin at 18h APF (*de novo* 10h), then at 20h APF (*de novo* 8h) and at 22h APF (*de novo* 6h). The image in Fig. 5B referred to by Reviewer 1, is of a wing where Fz is induced *de novo* at 22 h APF. In these wings, as expected, the core proteins localise asymmetrically in stereotypical swirling patterns throughout the wing surface (see Fig. 2M and also Strutt and Strutt, 2002; Strutt and Strutt 2007), but – usefully for our experiments – they broadly localise along the proximal-distal axis in the region analysed in Fig. 5B. Given the strong swirling in surrounding regions when inducing at >20h APF, we feel reasonably confident in assuming that the pattern is not due to a proximodistal cue present in the proximal wing. We appreciate that the original manuscript did not show images including the trichome pattern in adjacent regions, so this point would not have been clear, but we now include these in Supplementary Fig.S5. We have also added a note in the legend to Fig. 5B to clarify that the proximodistal pattern seen is local to this wing region.

Changes to manuscript: Text extended in Introduction and Results to better explain why we believe the *de novo* conditions that we use most likely result in a polarity pattern that is not significantly influenced by ‘global cues’. Now show zoomed-out images of the surrounding region around the experiment region proximal to the anterior cross-vein region in adult wings, showing that the polarity pattern does not become more proximodistal when induction time is longer, and also that there is not overall proximodistal polarity in proximal regions of the wing, arguing against an unknown proximodistal polarity cue at these stages of development (Fig.S5B-E’’’).

(2) The putative non-local cell scale signal must be more precisely defined (maybe also given a better name). It is not clear to me that one can separate cell-scale from molecular-scale signal.Local signals can redistribute within a cell (or membrane) so local signals are also cell-scale. Without a clear definition, it is difficult to interpret the results of the gene dosage experiments. The link between gene dosage and cell-scale signal is not rigorously stated. Related to this, the concluding statement of the introduction is too cryptic.

We thank the reviewer for raising this, as again a similar comment was made by Reviewer 1, so we are clearly falling short in defining the term. We have now had another attempt in the Introduction.

To more specifically answer the point made by the reviewer regarding molecular vs cellular, we are essentially being guided here by the prior computational modelling work, as at the biological level the details are still being worked out. A specific class of previous models only allowed ‘signals’ between core proteins to act ‘locally’, meaning within a cell junction, and within the models there was no explicit mechanism by which proteins on other junctions could ‘detect’ the polarity of a neighbouring junction (e.g. Amonlirdviman et al., 2005; Le Garrec et al., 2006; Fischer et al., 2013). Other models implicitly or explicitly encode a mechanism by which cell junctions can be influenced by the polarity of other junctions (e.g. Meinhardt, 2007; Burak and Shraiman, 2009; Abley et al., 2013; Shadkhoo and Mani, 2019), for instance by diffusion of a factor produced by localisation of particular planar polarity proteins.

We agree with the reviewer that a cell-scale signal will depend on ‘molecules’ and thus could be called ‘molecular-scale’, but here by ‘molecular-scale’ we mean signals that at the range of the sizes of molecules i.e. nanometers, rather than cell-scale signals that act at the size of cells i.e. micrometers. A caveat to our definition is that we implicitly include interactions that occur locally on cell junctions (<1 µm range) within ‘molecular-scale’, but this is a shorter range than ‘cellular-scale’ which requires signals acting over the diameter of a cell (3-5 µm). Nevertheless, we think the concept of ‘molecular-scale’ vs ‘cell-scale’ is a helpful one in this context, and have attempted to address the issue through a more careful definition of the terms.

Changes to manuscript: Text revised in Introduction and legend to Fig.1 to more carefully define ‘cell-scale signalling’ and to distinguish it from ‘molecular-scale signalling’. Final sentence of Introduction also altered so we no longer cryptically speculate on the nature of the cell-scale signal but leave this to the Discussion.

Minor comments.Some of the (clever) genetic manipulation may need more details in the text. For example:- Need to specify if the hs-flp approach induces expression throughout the tissue.

We apologise for the lack of clarity. In all the experiments, the *hs-FLP* transgene is present in all cells, and heat-shock results in ubiquitous expression.

Changes to manuscript: We have clarified this in the Results and Materials and Methods.

- Need to specify in the text that in the unpolarised condition the tissue is both dsh and fz mutant.

The reviewer is of course correct and we have updated this point in the text. The full genotype for the unpolarised condition is: *w dsh1 hsFLP22/y*;; *Act>>fz-mKate2sfGFP*, *fzP21/fzP21* (see Table S1). So this line is mutant for *dsh* and *fz* with induced expression of Fz-mKate2sfGFP.

Changes to manuscript: We have clarified this in the relevant part of the Results.

- Need to specify in the text that the experiment illustrated in Fig 5 is with hh-gal4.

As noted by the reviewer, we continued to use the same *hh-GAL4* repolarisation paradigm as in Fig.4 and this info was in the legend to Fig.5 legend. However, we agree it is helpful to be explicit about this in the main text.

Changes to manuscript: We have added this to this section of the Results.

- Need to address a possible shortcoming of the hh experiment, that the AP boundary is a region of high tension.

It is true that the AP boundary is under high tension in the wing disc (e.g. Landsberg et al., 2009). But we are not aware of any evidence that this higher tension persists into the pupal wing. In separate studies we have labelled for Myosin II in pupal wings (Trinidad et al 2025 Curr Biol; Tan & Strutt 2025 Nature Comms), and as far as we have noticed have not seen preferentially higher levels on the AP boundary. We think if tension were higher, the cell boundaries would appear straighter than in surrounding cells (as seen in the wing disc) and this is not evident in our images.

- Need to dispel the possibility that there is no residual polarisation (e.g. of other components) in fz1 mutant (I assume this is the case).

We use the null allele *fz[P21]* through this work, and we and others have consistently reported a complete loss of polarisation of other core proteins or downstream components in this background. The caveat to this is that core proteins that persist at cell junctions always appear at least slightly punctate in mutant backgrounds for other core proteins, and so any automated detection algorithm will always find evidence of individual cell polarity above a baseline level of uniform distribution. Hence we tend to use lack of local coordination of polarity (variance of cell polarity angle) as an additional measure of loss of polarisation, in addition to direct measures of average cell polarity. (We discuss this in the QuantifyPolarity manuscript Tan et al 2021 e.g. Fig.S6).

Changes to manuscript: We now include in the Materials and Methods section ‘Fly genetics…’ a much more extensive explanation of the evidence for specific mutant alleles being ‘null’ for planar polarity function (including *dsh1* as raised by Reviewer 1), specifically that they result in no detectable planar polarisation of either other core proteins or downstream effectors, and added appropriate references.

- Need to provide evidence that 50% gene dosage commensurately affect protein level.

This is a good suggestion. In the case of Stbm, we have already published a western blot showing that a reduction in gene dosage results in reduced protein levels (Strutt et al 2016, Fig.S6). We have now performed western blots to quantify protein levels upon reduction of *fmi*, *pk* and *dgo* levels (we actually used *EGFP-dgo* for the latter, as we don’t have antibodies that can detect endogenous Dgo on western blots).

Changes to manuscript: When presenting the dosage reduction experiments, we now refer back to Strutt et al., 2016 explicitly for Stbm, and have added western blot data for Fmi, Pk and EGFPDgo in new Fig.S2.

- I am surprised that the relationship with microtubule polarity was never investigated. Is this true?

We agree this is a point that needed further clarification, as Reviewer 1 made a related point regarding the two possible roles for microtubules, one being as a mediator of a global cue upstream of the core pathway, and the second (which we investigate in this manuscript) as a mediator of a cell-scale signal downstream of the core pathway.

Both the Uemura and Axelrod groups have published on potential upstream function as a global cue mediator in the *Drosophila* wing (e.g. Shimada et al., 2006; Harumoto et al., 2010; Matis et al., 2014).

Both groups have also looked out whether core pathway components could affect orientation of microtubules (Harumoto et al., 2010; Olofsson at al., 2014; Sharp and Axelrod 2016). Notably Harumoto et al., 2010 observed that in 24h APF wings, loss of Fz or Stbm did not alter microtubule polarity from a proximodistal orientation consistent with the microtubules aligning along the long cell axis in the absence of other cues. However, this did not rule out an instructive effect of Fz or Stbm on microtubule polarity during core pathway cell-scale signalling. The Axelrod lab manuscripts saw interesting effects of Pk protein isoforms on microtubule polarity, albeit not throughout the entire wing, which hinted at a potential role in cell-scale signalling. Taken together this prior work was the motivation for our directed experiments to specifically test whether the core pathway might generate cell-scale polarity by instructing microtubule polarity.

Changes to manuscript: We have revised the Results section ‘Microtubules do not…’ to make a clearer distinction regarding possible ‘upstream’ and ‘downstream’ roles of microtubules in *Drosophila* core pathway planar polarity and the motivation for our experiments investigating the latter.

- The authors suggest that polarity does not propagate as a wave. And yet the range measured in adult is longer than in the pupal wing. Explain.

Again an excellent point, also made by Reviewer 1, which we have now addressed explicitly in the manuscript. For the convenience of this reviewer, we lay out the reasons why we think the propagation of polarity seen in the adult is further than seen for core protein localisation.

There are three reasons why we might expect adult trichomes to show a different effect from the measured core protein polarity pattern seen in our experiments:

(i) we are assaying core protein polarity at 28h APF, but trichomes emerge at >32h APF, so there is still time for polarity to propagate a bit further from the boundary. We now have added data showing that by the point of trichome initiation, the wave of polarisation extends 3-4 cell rows (Fig.S5A).

(ii) it has long been known that a strong localisation of core proteins at a cell edge is not required for polarisation of trichome polarity from a boundary. For instance, in Strutt & Strutt 2007 we show clones of cells overexpressing Fz causing propagation through pk[pk-sple] mutant tissue where there is no detectable core protein polarity. We were following up prior observations of Adler et al 2000 in the wing and Lawrence et al 2004 in the abdomen.

(iii) there is evidence to suggest that the polarity of adult trichomes is locally coupled, possibly mechanically. This point is hard to prove without live imaging taking in both initial core protein localisation, the site of actin-rich trichome initiation and then the final orientation of the much larger microtubule filled trichome, and we’re not aware that such data exist. However, Wong & Adler 1993 (JCB) showed that over a number of hours trichomes become much larger and move towards the centre of the cell, presumably becoming decoupled from any core protein cue. The images in Guild … & Tilney, 2005 (MBoC) are also interesting to look at in this regard. Finally, septate junction proteins have been implicated in local alignment of trichomes, independently of the core pathway (Venema … & Auld, 2004 Dev Biol).

Changes to manuscript: Added new data in Fig.S5A showing where trichomes initiate under 6h *de novo* induction conditions, for comparison to core protein localisation and adult trichome data in Fig.5. Added some text explaining why adult trichome repolarisation might be stronger than the observed effects on core protein localisation in Discussion.

- The discussion states that the cell-intrinsic system remains to be fully characterised, implying that it has been partially characterised. What do we know about it?

As the reviewer probably realises, we were attempting to side-step a long speculative discussion about the various hints and ideas in the literature by grouping them under the umbrella of ‘remaining to be fully characterised’. We would argue that this current manuscript is the first to attempt to systematically investigate the nature of ‘cell-scale signalling’. The lack of prior work is probably due to two factors (i) pioneering theoretical work showed that a sufficiently strong global signal coupled with ‘local’ (i.e. confined to one cell junction) protein interactions was sufficient to polarise cells without the need to invoke the existence of a cell-scale signal; (ii) there is no easy way to identify cell-scale signals as their loss results in loss of polarity which will also occur if other (i.e. more locally acting) core pathway functions are compromised.

The main investigation of the potential for cell-scale signalling has been another set of theory studies (Burak and Shraiman 2009; Abley et al., 2013; Shadkhoo and Mani 2019) which have considered the possibility of diffusible signals. In our present work we have further considered the possibility of a ‘depletion’ model, based on the pioneering theory work of Hans Meinhardt, and as discussed above the possibility that microtubules could mediate a cell-scale signal.

Changes to manuscript: We have revised the Discussion to hopefully be clearer about the current state of knowledge.

**Reviewer #3:**
[…] Major commentsThe data are clearly presented and the manuscript is well written. The conclusions are well supported by the data.(1) The authors use a system to de novo establish PCP, which has the advantage of excluding global cues orienting PCP and thus to focus on the cell-intrinsic mechanisms. At the same time, the system has the limitation that it is unclear to what extent de novo PCP establishment reflects 'normal' cell scale PCP establishment, in particular because the Gal4/UAS expression system that is used to induce Fz expression will likely result in much higher Fz levels compared with the endogenous levels. The authors should briefly discuss this limitation.

We apologise if this wasn’t clear. We only used *GAL4/UAS* overexpression when we were generating an artificial boundary of Fz expression with *hh-GAL4* to induce repolarisation. The *de novo* induction system involves Fz::mKate2-sfGFP being expressed directly under an *Act5C* promoter without use of *GAL4/UAS*. In response to a comment from Reviewer 1 we have now carried out western blot analysis which shows that Fz::mKate2-sfGFP levels under *Act5C* are actually lower than endogenous Fz levels. As we achieve normal levels of polarity, similar to what we measure in wild-type conditions when measured using QuantifyPolarity, we assume that therefore Fz levels are not limiting under these conditions. However, we note that lower than normal levels of Fz might sensitise the system to perturbation, which in fact would be advantageous in our study, as it might for instance have been expected to more readily reveal dosage sensitivity of other components.

Changes to manuscript: We now describe the levels of expression achieved using the *de novo* induction system (Fig.S1C-D) and discuss possible consequences in the relevant Results sections and Discussion.

(2) Fig. 3. The authors use heterozygous mutant backgrounds to test the robustness of de novo PCP establishment towards (partial) depletion in core PCP proteins. The authors conclude that de novo polarization is 'extremely robust to variation in protein level'. Since the authors (presumably) lowered protein levels by 50%, this conclusion appears to be somewhat overstated. The authors should tune down their conclusion.

Reviewer 1 makes a similar point about whether we can argue that the lack of sensitivity to a 50% reduction in protein levels actually rules out the depletion model. To address the comments of both reviewers we had now added some further narrative and caveats in the text.

We nevertheless believe that the experiments shown effectively make the point that there is no strong dosage sensitivity – and it remains our contention that if protein levels were the key to setting up cell-scale polarity, then a 50% reduction would be expected to show an effect on the rate of polarisation. We further note that as Fz::mKate2-sfGFP levels are lower than endogenous Fz levels, the system might be expected to be sensitised to further dosage reductions, and despite this we fail to see an effect on rate of polarisation.

In a similar vein, Reviewer 2 requested data on whether dosage reduction altered protein levels by the expected amount. We have now added further explanation/references and western blot data to address this.

Changes to manuscript: Added some narrative and caveats regarding whether lowering levels more than 50% would add to our findings in the Discussion. Revised conclusions to be more cautious including altering section title to read ‘Planar polarity establishment is not highly sensitive to variation in protein levels of core complex components.

Also added westerns and text/references showing that for the tested proteins there is a reduction in protein levels upon removal of one gene dosage in Results section ‘Planar polarity establishment is…’ and Fig.S2.

Minor comments :(1) Page 3. The authors mention and reference that they used the PCA method to quantify cell polarity magnification and magnitude. It would help the unfamiliar reader, if the authors would briefly describe the principle of this method.

Changes to manuscript: More details have been added in Materials & Methods.

Significance:The manuscript contributes to our understanding of how planar cell polarity is established. It extends previous work by the authors (Strutt and Strutt, 2002,2007) that already showed that induction of core PCP pathway activity by itself is sufficient to induce de novo PCP. This manuscript further explores the underlying mechanisms. The authors test whether de novo PCP establishment depends on an 'inhibitory signal', as previously postulated (Meinhardt, 2007), but do not find evidence. They also test whether core PCP proteins help to orient microtubules (which could enhance cell intrinsic polarization of core PCP proteins), but, again, do not find evidence, corroborating previous work (Harumoto et al, 2010). The most significant finding of this manuscript, perhaps, is the observation that local de novo PCP establishment does not propagate far through the tissue. A limitation of the study is that the mechanisms establishing intrinsic cell scale polarity remain unknown. The work will likely be of interest to specialists in the field of PCP.